# Fractional-Order Control of Grid-Connected Photovoltaic System Based on Synergetic and Sliding Mode Controllers

Marcel Nicola [1],* and Claudiu-Ionel Nicola [1,2],*

1 Research and Development Department, National Institute for Research, Development and Testing in Electrical Engineering—ICMET Craiova, 200746 Craiova, Romania
2 Department of Automatic Control and Electronics, University of Craiova, 200585 Craiova, Romania
* Correspondence: marcel_nicola@yahoo.com or marcel_nicola@icmet.ro (M.N.); claudiu@automation.ucv.ro (C.-I.N.)

**Abstract:** Starting with the problem of connecting the photovoltaic (PV) system to the main grid, this article presents the control of a grid-connected PV system using fractional-order (FO) sliding mode control (SMC) and FO-synergetic controllers. The article presents the mathematical model of a PV system connected to the main grid together with the chain of intermediate elements and their control systems. To obtain a control system with superior performance, the robustness and superior performance of an SMC-type controller for the control of the $u_{dc}$ voltage in the DC intermediate circuit are combined with the advantages provided by the flexibility of using synergetic control for the control of currents $i_d$ and $i_q$. In addition, these control techniques are suitable for the control of nonlinear systems, and it is not necessary to linearize the controlled system around a static operating point; thus, the control system achieved is robust to parametric variations and provides the required static and dynamic performance. Further, by approaching the synthesis of these controllers using the fractional calculus for integration operators and differentiation operators, this article proposes a control system based on an FO-SMC controller combined with FO-synergetic controllers. The validation of the synthesis of the proposed control system is achieved through numerical simulations performed in Matlab/Simulink and by comparing it with a benchmark for the control of a grid-connected PV system implemented in Matlab/Simulink. Superior results of the proposed control system are obtained compared to other types of control algorithms.

**Keywords:** photovoltaic system; grid; sliding mode control; synergetic control; fractional-order control

## 1. Introduction

It is a well-known fact that it is important to use, to an increasing extent, renewable energy, characterized by the fact that it is generated from easily renewable sources which can thus be considered unlimited energy. Among these types of renewable energy sources, we refer to solar energy, wind energy, water energy, geothermal energy, etc. There is also a strong upward trend in the study and use of the PV system technology. Obviously, the study of its control systems was also developed in parallel with it [1].

Moreover, among the general approaches to the study of microgrid systems control, we can mention the study of the transient stability of a hybrid microgrid [2,3], the use of algorithms to offset lagging in the microgrid system [4], the optimization of the charging process of microgrid batteries [5,6], the study of the topologies of the converters used in the microgrid [7], the parallel coupling of the inverters in a microgrid [8], problems related to fault tolerance in the microgrid [9], and also problems related to the multi-grid dispatching in view of obtaining an economic optimum [10–14].

An inherent problem that arises is the control of the process of the PV system connection to a main grid. The components which ensure the connection of the PV system to the main grid are the DC-DC boost converter, the DC intermediate circuit, the DC-AC converter, the filtering block, and the transformer for the connection to the main grid, together with

their control systems. Further, a task of maximum importance in terms of the control of the grid-connected PV system is to maintain the DC voltage in the intermediate circuit—DC link voltage—$u_{dc}$ as precisely as possible. The control of this voltage is performed by a cascade control system in which the outer control loop controls the level of $u_{dc}$ voltage, and the inner control loops control currents $i_d$ and $i_q$ in the $dq$ rotating reference frame. Usually, the synchronization with the main grid is performed through a phase-locked loop (PLL), and the controllers of the control loops are of the classic proportional integrator (PI) type [15].

In order to obtain a more precise control of the voltage $u_{dc}$, we can use more complex control algorithms of the adaptive [16], robust [17,18], and predictive [19] types, but also intelligent control algorithms, such as fuzzy [20], neuro-fuzzy [21], genetics [22], particle swarm optimization (PSO) [23], and reinforcement learning [24].

A type of special controller used for the control of linear and nonlinear systems is based on PI and passivity theory [25]. This type of control, known at the beginning as the hyperstability control theory, is based on the appropriate description of the system in the closed loop in the form of the Lagrangian or the port-controlled Hamiltonian, which defines the behavior of the system through energy functions. This description is generally expressed in the form of solutions to partial differential equations, which require an increased degree of difficulty in the implementation of these types of controllers.

An alternative for the synthesis of some controllers for nonlinear systems, which ensures the parametric robustness and maintains a low order of the synthesized control law, is the use of SMC [26]. Among the disadvantages of the SMC-type control, we mention the occurrence of the chattering phenomenon, which represents the occurrence of oscillations in the control input due to the process of synthesis of the controller. For this purpose, to reduce oscillations, transition functions smoother than the sgn function are used (between conventional thresholds +1 and −1) followed by a corresponding filtration. Further, a type of controller suitable for the control of nonlinear systems is the synergetic controller [27]. It is important to recall that, by using such a controller, it is not necessary to linearize the nonlinear model around a static operating point because this type of controller provides good performance for the entire operating range of the nonlinear model.

By adding the FO calculus and the fractional differentiation and integration operators [28,29], control laws can be obtained with a higher degree of refinement due to the additional control parameter which represents the order of the fractional differentiation and integration operator. Thus, a control structure in which classical PI controllers are replaced with FO-synergetic controllers is presented in [30]. Superior results are obtained by using the control structure proposed in this article, as well as the proposed macro-variables for the synthesis of control laws.

The main contributions of this article consist in replacing the classic PI-type controllers in the control loops of $u_{dc}$ voltage and $i_d$ and $i_q$ currents with, respectively, the FO-SMC and FO-synergetic type controllers. Thus, the synthesis of the control laws related to these types of controllers is presented, as well as the results obtained by numerical simulations in Matlab/Simulink for the control of the grid-connected PV system, in which classic PI, synergetic, or FO-synergetic controllers are used for the inner control loops of currents $i_d$ and $i_q$, and classic PI, SMC, or FO-SMC controllers are used for the outer control loop of $u_{dc}$. Owing to the levels of freedom and the refinement brought by the fractional calculus for the SMC and synergetic algorithms, the performances of the control system will be superior to those presented in the benchmark implementation in [15] but will also compare to the results obtained in other papers using the basic approach presented in [15].

The main contributions of this paper can be summarized as follows:

- We propose the cascade structure of the control system of the grid-connected PV system based on the robustness of an SMC-type controller for the control of $u_{dc}$ voltage in the DC intermediate circuit, combined with the flexibility of using synergetic control for the control of currents $i_d$ and $i_q$.

- The synthesis of the control laws by SMC- and synergetic-type controllers using the fractional calculus for integration operators and differentiation operators.
- We realized the numerical simulations in Matlab/Simulink, and by comparing them with a benchmark for the control of a grid-connected PV system implemented in Matlab/Simulink, superior results of the proposed control system compared to other types of control algorithms are presented.

The other sections of the paper are structured as follows: Section 2 presents a mathematical model of the grid-connected PV system. Section 3 presents the control of the grid-connected PV system using the FO calculus for the SMC controller and synergetic controllers. Section 4 presents the numerical simulations in Matlab/Simulink for the control of the grid-connected PV system using FO-SMC and FO-synergetic controllers and the analysis thereof, and some conclusions are presented in Section 5.

## 2. Mathematical Model of the Grid-Connected PV System

Following [15,27,31], which show the mathematical model of a grid-connected PV system, Figure 1 shows the general diagram of such a system. The PV array system is modeled in [15,27,31] and the input parameters are radiation and temperature. The list of component blocks additionally includes a DC boost converter along with the maximum power point tracking (MPPT) module and a three-phase DC-AC converter. The notations in Figure 1 are the usual ones.

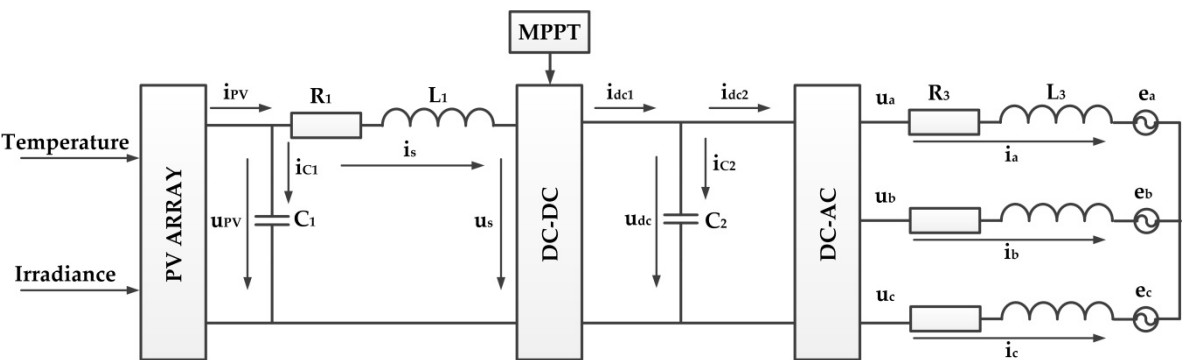

**Figure 1.** Block diagram of the main circuit diagram of the grid-connected PV system.

Thus, the following equations can be written to describe the operation of the grid-connected PV system:

$$C_1 \frac{du_{PV}}{dt} = i_{PV} - i_s \tag{1}$$

$$u_{PV} = R_1 i_s + L_1 \frac{di_s}{dt} + u_s \tag{2}$$

$$C_2 \frac{du_{dc}}{dt} = i_{dc1} - i_{dc2} \tag{3}$$

$$u_{abc} - e_{abc} = R_3 i_{abc} + L_3 \frac{di_{abc}}{dt} \tag{4}$$

where $u_{abc}$ represents the output voltage of the DC-AC (voltage source converter—VSC) converter with $u_{abc} = \begin{bmatrix} u_a & u_b & u_c \end{bmatrix}^T$, $e_{abc}$ represents the grid voltage with $e_{abc} = \begin{bmatrix} e_a & e_b & e_c \end{bmatrix}^T$, and $i_{abc}$ represents the alternating current with $i_{abc} = \begin{bmatrix} i_a & i_b & i_c \end{bmatrix}^T$.

Park's transformation based on the *P* matrix is well known:

$$P = \begin{bmatrix} \sin(\omega t) & \sin\left(\omega t - \frac{2\pi}{3}\right) & \sin\left(\omega t + \frac{2\pi}{3}\right) \\ \cos(\omega t) & \cos\left(\omega t - \frac{2\pi}{3}\right) & \cos\left(\omega t + \frac{2\pi}{3}\right) \\ \frac{1}{2} & \frac{1}{2} & \frac{1}{2} \end{bmatrix} \tag{5}$$

By applying this transformation to the *abc* reference system, we obtain the usual quantities in Figure 1 in the *dq* reference system ($u_{dq0} = Pu_{abc}$, $e_{dq0} = Pe_{abc}$, $i_{dq0} = Pi_{abc}$). Thus, Equation (4) becomes

$$u_{dq0} - e_{dq0} = R_3 i_{dq0} + L_3 \frac{di_{dq0}}{dt} + L_3 \begin{bmatrix} -\omega i_q \\ \omega i_d \\ 0 \end{bmatrix} \tag{6}$$

By components, this equation can be rewritten in the form of Equations (7) and (8):

$$L_3 \frac{di_d}{dt} = -R_3 i_d + \omega L_3 i_q - e_d + u_d = u_{3d} + u_d \tag{7}$$

$$L_3 \frac{di_q}{dt} = -R_3 i_q - \omega L_3 i_d - e_d + u_q = u_{3q} + u_q \tag{8}$$

where $u_{id}$ and $u_{iq}$ represent the control variables for the command of the DC-AC converter (which is of voltage source converter—VSC-type). Equations (7) and (8) include the following notations: $u_{3d} = -R_3 i_d + \omega L_3 i_q - e_d$ and $u_{3q} = -R_3 i_q - \omega L_3 i_d - e_q$.

The MPPT algorithm is presented in [15,26,29]; it will provide the duty cycle signal (*D*) to control the DC boost converter. Thus, the following relations can be written:

$$i_{dc1} = (1 - D)i_s \tag{9}$$

$$u_s = (1 - D)u_{dc} \tag{10}$$

## 3. Control of the Grid-Connected PV System

The control of a grid-connected PV system is presented at length in [15,27,31], both in normal operation and in low-voltage ride through (LVRT). The controllers used for the inner control loops of currents $i_d$ and $i_q$ are of the classic PI type or synergetic type [27], while the controller for the outer control loop of $u_{dc}$ is of the classic PI type [15,31]. Figure 2 shows the control scheme for the connection of a PV system to the power grid under normal operation. The control loops of currents $i_d$ and $i_q$ and of $u_{dc}$ are presented schematically in Figure 3. In this section, we will present several basic elements of the FO calculus to synthesize the fractional-type control laws in the case of using the design and synthesis procedures of the SMC and synergetic controllers.

### 3.1. Notions and Notations for Fractional-Order Calculus

To achieve a refinement of the differential and integral calculus, the non-integer order operator is added as $aD_t^\alpha$, where the FO is noted with $\alpha$, and the limits of the use of the operator are denoted *a* and *t* [28,29].

$$aD_t^\alpha = \begin{cases} \frac{d^\alpha}{dt^\alpha} & \text{Re}(\alpha) > 0 \\ 1 & \text{Re}(\alpha) = 0 \\ \int_a^t (dt)^{-\alpha} & \text{Re}(\alpha) < 0 \end{cases} \tag{11}$$

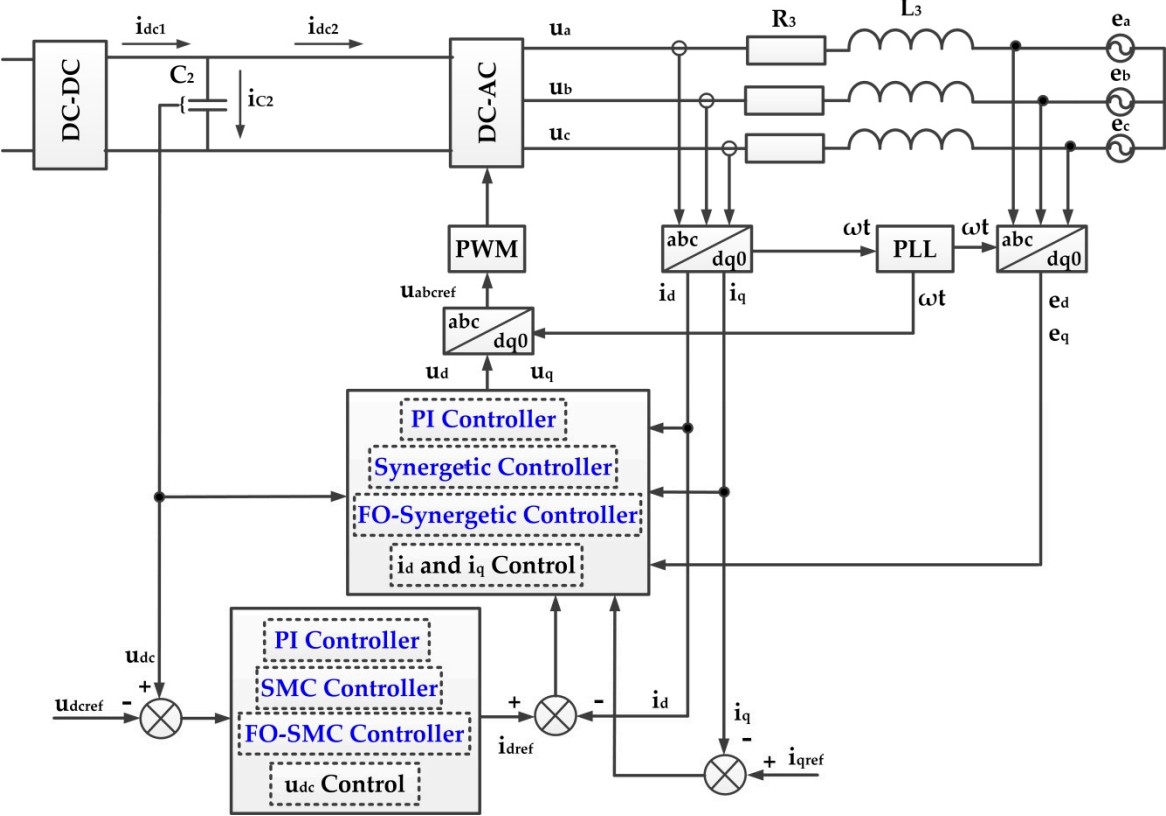

**Figure 2.** Block diagram of fractional-order sliding mode control (FO-SMC) and FO-synergetic control of the grid-connected PV system.

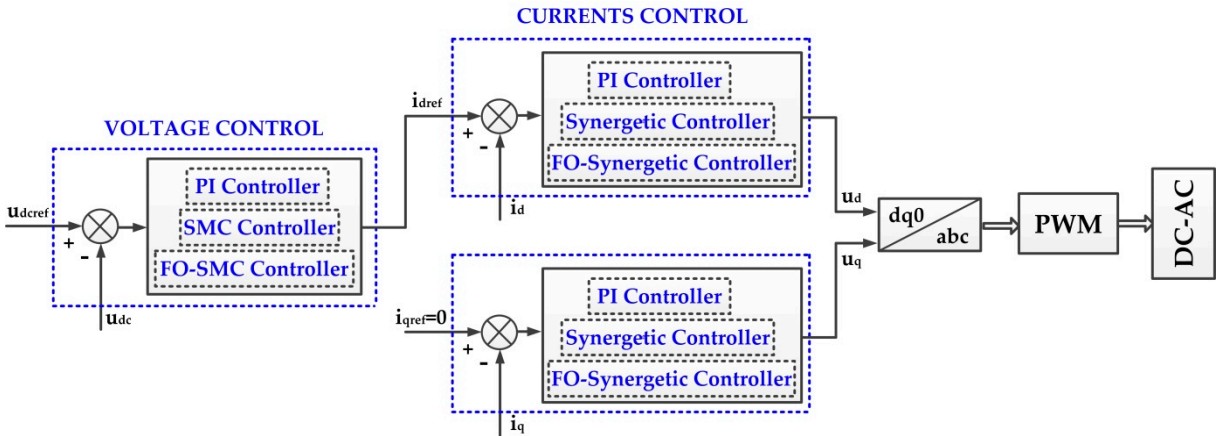

**Figure 3.** General scheme for cascade control of the grid-connected PV system.

Further, a common alternative definition is given by the Riemann–Liouville differintegral [28,29]:

$$aD_t^\alpha f(t) = \frac{1}{\Gamma(m-\alpha)} \left(\frac{d}{dt}\right)^m \int_\alpha^t \frac{f(\tau)}{(t-\tau)^{\alpha-m+1}} d\tau \tag{12}$$

where $m-1 < \alpha < m$, $m \in N$, and $\Gamma(\cdot)$ is Euler's gamma function.

For the practical implementation by numerical calculation, the Grünwald–Letnikov definition is presented as follows [28,29]:

$$aD_t^\alpha f(t) = \lim_{h \to 0} \frac{1}{h^\alpha} \sum_{j=0}^{\left(\frac{t-\alpha}{h}\right)} (-1)^j \begin{pmatrix} \alpha \\ j \end{pmatrix} f(t - jh) \tag{13}$$

where $(\cdot)$ is the integer part.

The Laplace transform can also be applied in the non-integer case similarly to the integer case (in terms of the power of the complex operator $s$). A special case is when the power $\alpha$ of operator $s$ is of the commensurate order type $q$, $(q \in R^+, 0 < q < 1, \alpha_k = kq)$. For $\lambda = s^q$, the transfer function $H(\lambda)$ can be written as

$$H(\lambda) = \frac{\sum\limits_{k=0}^{m} b_k \lambda^k}{\sum\limits_{k=0}^{n} a_k \lambda^k} \tag{14}$$

For the numerical implementations in embedded systems in real time, it is important to emphasize that the results of the fractional calculus cannot be implemented directly, but an integer-order approximation of these calculi is used on a specified frequency range $(\omega_b, \omega_h)$, by using Oustaloup recursive filters.

For $s^\gamma$ with $0 < \gamma < 1$, an approximation can be used as follows [28,29]:

$$G_f(s) = K \prod_{k=-N}^{N} \frac{s + \omega_k'}{s + \omega_k} \tag{15}$$

where $\omega_k'$, $\omega_k$, and $K$ are given by

$$\omega_k' = \omega_b \left(\frac{\omega_h}{\omega_b}\right)^{\frac{k+N+\frac{1}{2}(1-\gamma)}{2N+1}} ; \; \omega_k = \omega_b \left(\frac{\omega_h}{\omega_b}\right)^{\frac{k+N+\frac{1}{2}(1+\gamma)}{2N+1}} ; \; k = \omega_h^\gamma \tag{16}$$

A refined form of Oustaloup-type filters is given by the following relations [28]:

$$s^\alpha \approx \left(\frac{d\omega_h}{b}\right)^\alpha \left(\frac{ds^2 + b\omega_h s}{d(1-\alpha)s^2 + b\omega_h s + d\alpha}\right) G_p \tag{17}$$

$$G_p = K \prod_{k=-N}^{N} \frac{s + \omega_k'}{s + \omega_k}; \; \omega_k = \left(\frac{b\omega_h}{d}\right)^{\frac{\alpha+2k}{2N+1}}; \; \omega_k' = \left(\frac{d\omega_b}{b}\right)^{\frac{\alpha-2k}{2N+1}} \tag{18}$$

where usually parameters $b = 10$ and $d = 9$.

### 3.2. Fractional-Order Sliding Mode Control

Starting from Equation (3) and using $S_a$, $S_b$, and $S_c$ to denote the switching function for the DC-AC converter in Figure 1, the following equation is obtained in the *abc* frame:

$$C_2 \frac{du_{dc}}{dt} = i_{dc1} - (i_a S_a + i_b S_b + i_c S_c) \tag{19}$$

The switching functions $Sd$ and $Sq$ can be obtained by using transformation (5):

$$\begin{bmatrix} S_d & S_q & 0 \end{bmatrix}^T = P \begin{bmatrix} S_a & S_b & S_c \end{bmatrix}^T \tag{20}$$

Based on these, Equation (19) becomes

$$C_2 \frac{du_{dc}}{dt} = i_{dc1} - \frac{3}{2}\left(i_d S_d + i_q S_q\right) \tag{21}$$

In Equation (21), $i_{dc1}$ is given by the relations (9) and (10) which depend on the DC boost converter and the MPPT algorithm, which we will consider as an optimized form given by [15,31], so we will consider that it is necessary for the other terms of the right member to be calculated by the sliding mode control technique to maintain $u_{dc}$ at a prescribed value $u_{dcref}$ (which is considered constant or has slow variations relative to the variation of the other quantities in the control system). Additionally, in [26], it is demonstrated that, when the three-phase grid system is symmetrical, $i_d$ represents the direct current and reference $i_{qref} = 0$ is selected, and thus Equation (21) becomes:

$$C_2 \frac{du_{dc}}{dt} = i_{dc1} - \frac{3}{2} i_{dref} S_d \tag{22}$$

Following the sliding mode control design procedure, the reference $i_{dref}$ for the inner control loop of currents $i_d$ and $i_q$ will be determined. Thus, we define the state variable $x_1$ as

$$x_1 = u_{dc} - u_{dcref} \tag{23}$$

We define the switching surface $S$:

$$\begin{cases} S = c_1 x_1 + x_2 \\ \dot{S} = c_1 x_2 + \dot{x}_2 \end{cases} \tag{24}$$

where the state variable $x_2$ is defined by:

$$x_2 = \dot{x}_1 = -\dot{u}_{dc} \tag{25}$$

To achieve convergence, the following is required:

$$\dot{S} = -\varepsilon \text{sgn} S - kS \tag{26}$$

where $\varepsilon$ and $k$ are positive constants.

From the calculation, we obtain:

$$\ddot{x}_1 = \dot{x}_2 = -\ddot{u}_{dc} = \frac{3}{2} \frac{S_d}{C_2} \dot{i}_{dref} - \frac{\dot{i}_{dc1}}{C_2}, \tag{27}$$

and thus the following can be written:

$$-\varepsilon \text{sgn} S - kS = c_1 x_2 + \frac{3}{2} \frac{1}{C_2} S_d \dot{i}_{dref} - \frac{\dot{i}_{dc1}}{C_2} \tag{28}$$

Following [32], to improve convergence and reduce high-frequency oscillations, the sgn function is replaced with the function below:

$$h(x) = \frac{2}{1 + e^{-a(x-b)}} - 1 \tag{29}$$

For $a = 4$ and $b = 0$, $h \in [-1 \ 1]$, and a smoothed transition is achieved between $-1$ and 1. From this, the output of the SMC-type controller can be inferred:

$$i_{dref} = \frac{2}{3} \frac{C_2}{S_d} \int_0^t \left[ -(c_1 x_2 + kS - \varepsilon h(S)) + \frac{\dot{i}_{dc1}}{C_2} \right] dt \tag{30}$$

For the fractional case, the switching surface is defined as:

$$S = c_1 x_1 + c_2 D^\mu x_1 = c_1 x_1 + c_2 D^{\mu-1} x_2, \tag{31}$$

where the fractional differential operator $D$ is defined in relation (11).

By calculating $\dot{S}$, we obtain:

$$\dot{S} = c_1 \dot{x}_1 + c_2 D^{\mu+1} x_1 = c_1 x_2 + c_2 D^{\mu-1} \dot{x}_2, \tag{32}$$

which can be rewritten using Equation (27):

$$\dot{S} = c_1 x_2 + c_2 D^{\mu-1} \left( \frac{3}{2} \frac{1}{C_2} S_d \dot{i}_{dref} - \frac{\dot{i}_{dc1}}{C_2} \right) \tag{33}$$

Using Equation (26), we obtain:

$$- \varepsilon h(S) - kS - c_1 x_2 = c_2 D^{\mu-1} \left( \frac{3}{2} \frac{1}{C_2} S_d \dot{i}_{dref} - \frac{\dot{i}_{dc1}}{C_2} \right) \tag{34}$$

By applying operator $D^{1-\mu}$ to relation (34), we obtain:

$$D^{1-\mu}(-\varepsilon h(S) - kS - c_1 x_2) = c_2 \frac{3}{2} \frac{1}{C_2} S_d \dot{i}_{dref} - c_2 \frac{\dot{i}_{dc1}}{C_2} \tag{35}$$

The output of the FO-SMC type controller can thus be inferred from the following:

$$i_{dref} = \frac{2}{3 c_2} \frac{C_2}{S_d} \int_0^t \left[ c_2 \frac{\dot{i}_{dc1}}{C_2} + D^{1-\mu}(-\varepsilon h(S) - kS - c_1 x_2) \right] dt \tag{36}$$

Both in Equation (30) and in Equation (36), in order to avoid the uncontrolled increase in $i_{dref}$ due to the zero-crossings of the signal $S_d$, it will be replaced in the practical implementation with $S'_d = S_d + c_3$, where $c_3 > 0$ becomes a new level of freedom in the design of the FO-SMC controller.

### 3.3. Fractional-Order Synergetic Control

It is well known that synergetic control can be considered as a generalization of sliding mode control. Thus, synergetic control can be applied to nonlinear systems described by the general form [27,32]

$$\dot{x} = f(x, u, t) \tag{37}$$

where $x$ represents the state vector, $x \in \Re^n$; $f(.)$ represents the continuous nonlinear function; u represents the control vector, $u \in \Re^m$, $(m < n)$.

The synergetic control procedure includes the selection of a macrovariable $\psi(x, t)$ which depends on the states of the system, for each control input. The system will be forced to evolve to manifolds $\psi = 0$, according to the following equation:

$$T\dot{\psi} + \psi = 0 \tag{38}$$

where $T > 0$ is selected to obtain the desired convergence rate.

By differentiating the macrovariable $\Psi$, we obtain:

$$\dot{\psi} = \frac{\partial \psi}{\partial x} \dot{x}, \tag{39}$$

and by inserting the relation (39) into Equation (38), we obtain:

$$T\frac{\partial \psi}{\partial x} \dot{x} + \psi = 0 \tag{40}$$

The explicit forms of $\dot{x}$ states in the mathematical model of the controlled system are inserted into Equation (40). This results in the control law as follows:

$$u = u(x, \psi(x, t), T, t) \tag{41}$$

Next, we will apply the integer-order and fractional-order synergetic control procedures to replace the classic PI-type control loops of currents $i_d$ and $i_q$. The outputs of the synergetic controller will be $u_d$ and $u_q$ (see Figure 2).

For the d-axis, for $k_d > 0$, we choose the macrovariable $\Psi_d$ as follows:

$$\psi_d = \left( u_{dcref} - u_{dc} \right) + k_d \left( i_{dref} - i_d \right) \tag{42}$$

We define another state variable $x_2$ (in addition to the state variable $x_1$ in Equation (23)):

$$\begin{cases} x_1 = u_{dcref} - u_{dc} \\ x_2 = i_{dref} - i_d \end{cases} \tag{43}$$

Based on the relation (43) for slow variations of the reference quantities or quasi-stationary regime, the following relation can be written:

$$\begin{cases} \dot{x}_1 = -\dot{u}_{dc} \\ \dot{x}_2 = -\dot{i}_d \end{cases} \tag{44}$$

Using these, we obtain the macrovariable derivative $\Psi_d$ defined in relation (42), of the following form:

$$\dot{\psi}_d = \dot{x}_1 + k_d \dot{x}_2 = -\dot{u}_{dc} - k_d \dot{i}_d \tag{45}$$

Based on these, for $T = T_1$, Equation (40) becomes:

$$T_1 \left( -\dot{u}_{dc} - k_d \dot{i}_d \right) + \left( u_{dcref} - u_{dc} \right) + k_d \left( i_{dref} - i_d \right) = 0 \tag{46}$$

Using Equation (7), Equation (46) becomes:

$$-T_1 \dot{u}_{dc} - T_1 k_d \frac{1}{L_3} (u_{3d} - u_d) + \left( u_{dcref} - u_{dc} \right) + k_d \left( i_{dref} - i_d \right) = 0 \tag{47}$$

After rearranging the terms in Equation (47), we can write:

$$T_1 k_d \frac{1}{L_3} u_d = -T_1 \dot{u}_{dc} - T_1 k_d \frac{1}{L_3} u_{3d} + \left( u_{dcref} - u_{dc} \right) + k_d \left( i_{dref} - i_d \right) \tag{48}$$

Based on this, we obtain the control law $u_d$ as follows:

$$u_d = \frac{L_3}{T_1 k_d} \left[ -T_1 \dot{u}_{dc} - T_1 k_d \frac{1}{L_3} u_{3d} + \left( u_{dcref} - u_{dc} \right) + k_d \left( i_{dref} - i_d \right) \right] \tag{49}$$

For axis $d$ in the fractional case, the macrovariable is chosen:

$$\psi_d = D^\mu x_1 + k_d x_2 \tag{50}$$

By deriving Equation (50), we obtain:

$$\dot{\psi}_d = D^\mu \dot{x}_1 + k_d \dot{x}_2 = -D^\mu \dot{u}_{dc} - k_d \dot{i}_d \tag{51}$$

Based on these, Equation (40) becomes

$$T_1 \left( -D^\mu \dot{u}_{dc} - k_d \dot{i}_d \right) + D^\mu \left( u_{dcref} - u_{dc} \right) + k_d \left( i_{dref} - i_d \right) = 0 \tag{52}$$

Using Equation (7), Equation (52) becomes:

$$-T_1 D^{\mu+1} u_{dc} - T_1 k_d \frac{1}{L_3}(u_{3d} + u_d) + D^\mu\left(u_{dcref} - u_{dc}\right) + k_d\left(i_{dref} - i_d\right) = 0 \qquad (53)$$

After rearranging the terms in Equation (53), we can write:

$$T_1 k_d \frac{1}{L_3} u_d = -T_1 D^{\mu+1} u_{dc} - T_1 k_d \frac{1}{L_3} u_{3d} + D^\mu\left(u_{dcref} - u_{dc}\right) + k_d\left(i_{dref} - i_d\right) \qquad (54)$$

Based on this, we obtain the control law $u_d$ as follows:

$$u_d = \frac{L_3}{T_1 k_d}\left[-T_d D^{\mu+1} u_{dc} - T_1 k_d \frac{1}{L_3} u_{3d} + D^\mu\left(u_{dcref} - u_{dc}\right) + k_d\left(i_{dref} - i_d\right)\right] \qquad (55)$$

For the $q$-axis, for $k_q > 0$, we choose the macrovariable $\Psi_q$ of the following form:

$$\psi_q = i_{qref} - i_q \qquad (56)$$

We define another state variable $x_3$ (in addition to the state variables $x_1$ and $x_2$ in Equation (43)):

$$\begin{cases} x_1 = u_{dcref} - u_{dc} \\ x_2 = i_{dref} - i_d \\ x_3 = i_{qref} - i_q \end{cases} \qquad (57)$$

Based on relation (57), because $i_{qref}$ is set to zero, we can write:

$$\begin{cases} \dot{x}_1 = -\dot{u}_{dc} \\ \dot{x}_2 = -\dot{i}_d \\ \dot{x}_3 = -\dot{i}_q \end{cases} \qquad (58)$$

Using these, we obtain the macrovariable derivative $\Psi_q$ defined in relation (56), of the following form:

$$\dot{\psi}_q = \dot{x}_3 \qquad (59)$$

Based on these, for $T = T_2$, Equation (40) becomes:

$$-T_2 \dot{i}_q + \left(i_{qref} - i_q\right) = 0 \qquad (60)$$

Using Equation (8), Equation (60) becomes:

$$-T_2 \frac{1}{L_3}\left(u_{3q} + u_q\right) + i_{qref} - i_q = 0 \qquad (61)$$

After rearranging the terms in Equation (61), we can write:

$$\left(u_{3q} + u_q\right) = \frac{L_3}{T_2}\left(i_{qref} - i_q\right) \qquad (62)$$

Based on this, we obtain the control law $u_q$ as follows:

$$u_q = \frac{L_3}{T_2}\left(i_{qref} - i_q\right) - u_{3q} \qquad (63)$$

For the $q$-axis in the fractional case, the macrovariable is chosen:

$$\psi_q = D^\mu x_3 + k_q \int_0^t x_3(t)dt \qquad (64)$$

By deriving Equation (64), we obtain:

$$\dot{\psi}_q = D^\mu \dot{x}_3 + k_q x_3 = -D^\mu \dot{i}_q + k_q \left( i_{qref} - i_q \right) \tag{65}$$

Based on these, Equation (40) becomes:

$$T_2 \left[ D^\mu \left( -\frac{1}{L_3} (u_{3q} + u_q) \right) + k_q \left( i_{qref} - i_q \right) \right] + D^\mu \left( i_{qref} - i_q \right) + k_q \int_0^t \left( i_{qref} - i_q \right) dt = 0 \tag{66}$$

By using Equation (8) and applying the fractional operator defined in relation (11) to both members of Equation (66), and considering that $D^{-\mu}$ becomes $I_\mu$, we can write:

$$-\frac{T_2}{L_3} (u_{3q} + u_q) + T_2 k_q I_\mu \left( i_{qref} - i_q \right) + \left( i_{qref} - i_q \right) + k_q I_{\mu+1} \left( i_{qref} - i_q \right) = 0 \tag{67}$$

After rearranging the terms in Equation (67), we can write:

$$\frac{T_2}{L_3} u_q = T_q k_q I_\mu \left( i_{qref} - i_q \right) + \left( i_{qref} - i_q \right) + k_q I_{\mu+1} \left( i_{qref} - i_q \right) - \frac{T_2}{L_3} u_{3q} \tag{68}$$

Based on this, we obtain the control law $u_q$ as follows:

$$u_q = \frac{L_3}{T_2} \left[ T_2 k_q I_\mu \left( i_{qref} - i_q \right) + \left( i_{qref} - i_q \right) + k_q I_{\mu+1} \left( i_{qref} - i_q \right) - \frac{T_2}{L_3} u_{3q} \right] \tag{69}$$

In the case of integer-order synergetic control, the control inputs $u_d$ and $u_q$ are provided by Equations (49) and (63), and in the case of the fractional synergetic control, the control inputs $u_d$ and $u_q$ are provided by Equations (55) and (69). By applying the inverse Park transform, the actual control inputs $u_{abc}$ (see Figure 2) are obtained as follows:

$$u_{abc} = P^{-1} u_{dq0} \tag{70}$$

## 4. Numerical Simulations and Analysis for the Control of the Grid-Connected PV System Using FO-SMC and FO-Synergetic Controllers

In this section, starting from the controllers synthesized in the previous section, we will present the obtained results of the global system for the control of the grid-connected PV system, in which classic PI, synergetic, or FO-synergetic controllers are used for the inner control loops of currents $i_d$ and $i_q$ and classic PI, SMC, or FO-SMC controllers are used for the outer control loop of $u_{dc}$. Owing to the levels of freedom and the refinement brought by the fractional calculus for the SMC and synergetic algorithms, it will be demonstrated, through numerical simulations, using the Matlab/Simulink environment, that superior performance is achieved using the proposed control system. The system described in Figures 2 and 3 is implemented in Matlab/Simulink and the block diagram is presented in Figure 4.

The presented implementation starts from an example in Matlab/Simulink [15], which presents the control system and the performances for a 100 kW model of the grid-connected PV array. The reference value of the voltage in the DC intermediate circuit is set to 500 V and the rated AC voltage supplied by the DC-AC converter is of 260 V. Further, the load is connected to the main grid through a distribution transformer with a rated voltage of 25 kV/260 V. The MPPT algorithm and its performances are presented and implemented in [15,31] and are used in the implementation presented in this article as a block function.

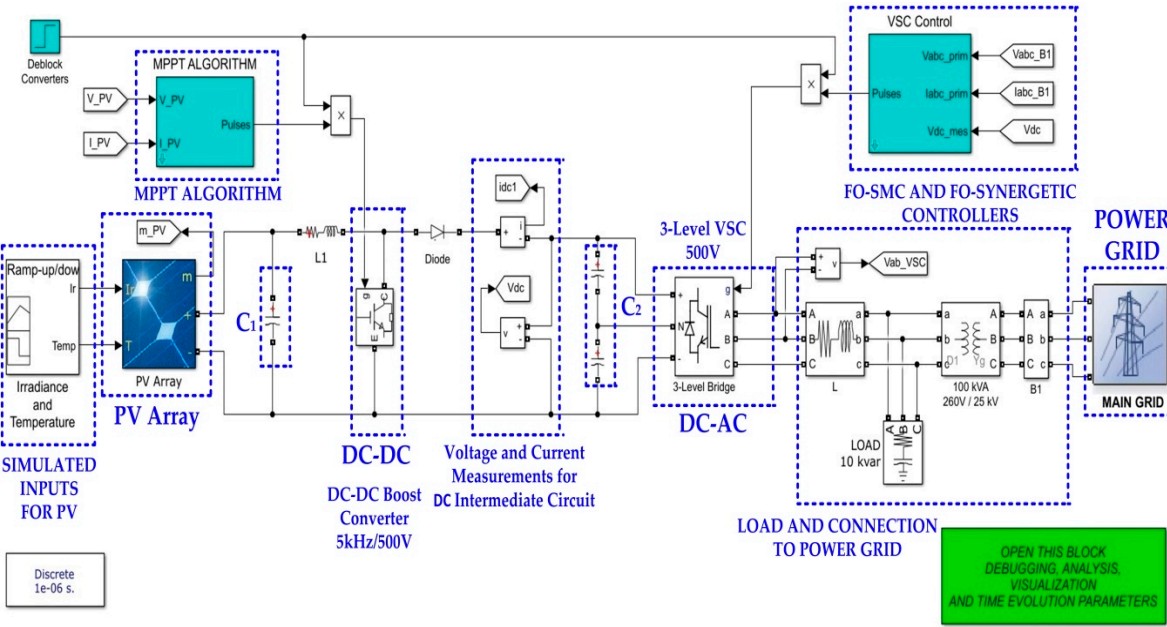

**Figure 4.** Matlab/Simulink implementation block diagram for control of the grid-connected PV system using FO-SMC and FO-synergetic controllers.

In order to have the smallest possible fluctuations when the DC-AC converter supplies a variable load, the importance of the precise control of the voltage level in the DC intermediate circuit—$u_{dc}$—is well known. For this, two cascade control loops are used, an outer one for the control of $u_{dc}$ and two inner loops for the control of currents $i_d$ and $i_q$. The current reference and $i_{dref}$ are supplied by the output of the outer loop controller, and $i_{qref}$ is set to zero [26].

Figures 5 and 6 show the Matlab/Simulink implementations of the control laws synthesized in Section 3 for the most complex case, where the controller of the outer control loop of voltage $u_{dc}$ is of the FO-SMC type, and the inner control loops of currents $i_d$ and $i_q$ are of the FO-synergetic type. Moreover, in Figure 4, the signal filtering at the DC-AC converter output is achieved using a 10 kvar bank capacitor, which can be considered as the load for the control system.

The operation of the PV array is also implemented in [15], and the time variation of the irradiance and temperature input signals is shown in Figure 7. The PV array includes 330 SunPower-type modules which can supply a maximum of 100.7 kW (305.2 W/modules), and each module is characterized by an open-circuit voltage of $V_{oc}$ = 64.2 V and a short-circuit current of $I_{sc}$ = 5.96 A. In the Matlab/Simulink implementation, the sample time used is of one microsecond for the Pulse Width Modulation (PWM) generator command signals for the DC-DC and DC-AC converters. For the control system of the voltage and currents, but also for the PLL-type synchronization loop, the sampling period is of 0.1 ms.

In the Matlab/Simulink implementation in [15], for the first 50 ms, the operation of the converters is bypassed during the period when the control system operates in the open loop. After the first 50 ms, the controllers come into operation both for the DC-DC converter and for the MPPT algorithm provided by [31], but also for the control of the DC-AC converter whose improved FO-SMC and FO-synergetic controllers proposed in this article provide superior performance compared to the classical PI controllers proposed in [15], both in stationary mode and in dynamic mode (see Figures 8–11).

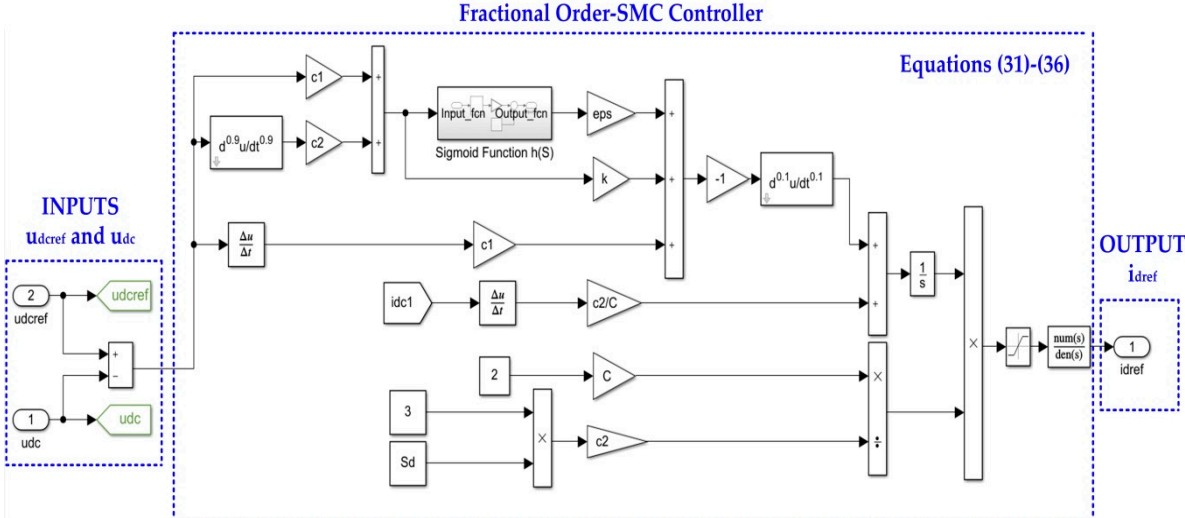

**Figure 5.** Matlab/Simulink implementation block diagram for the FO-SMC controller.

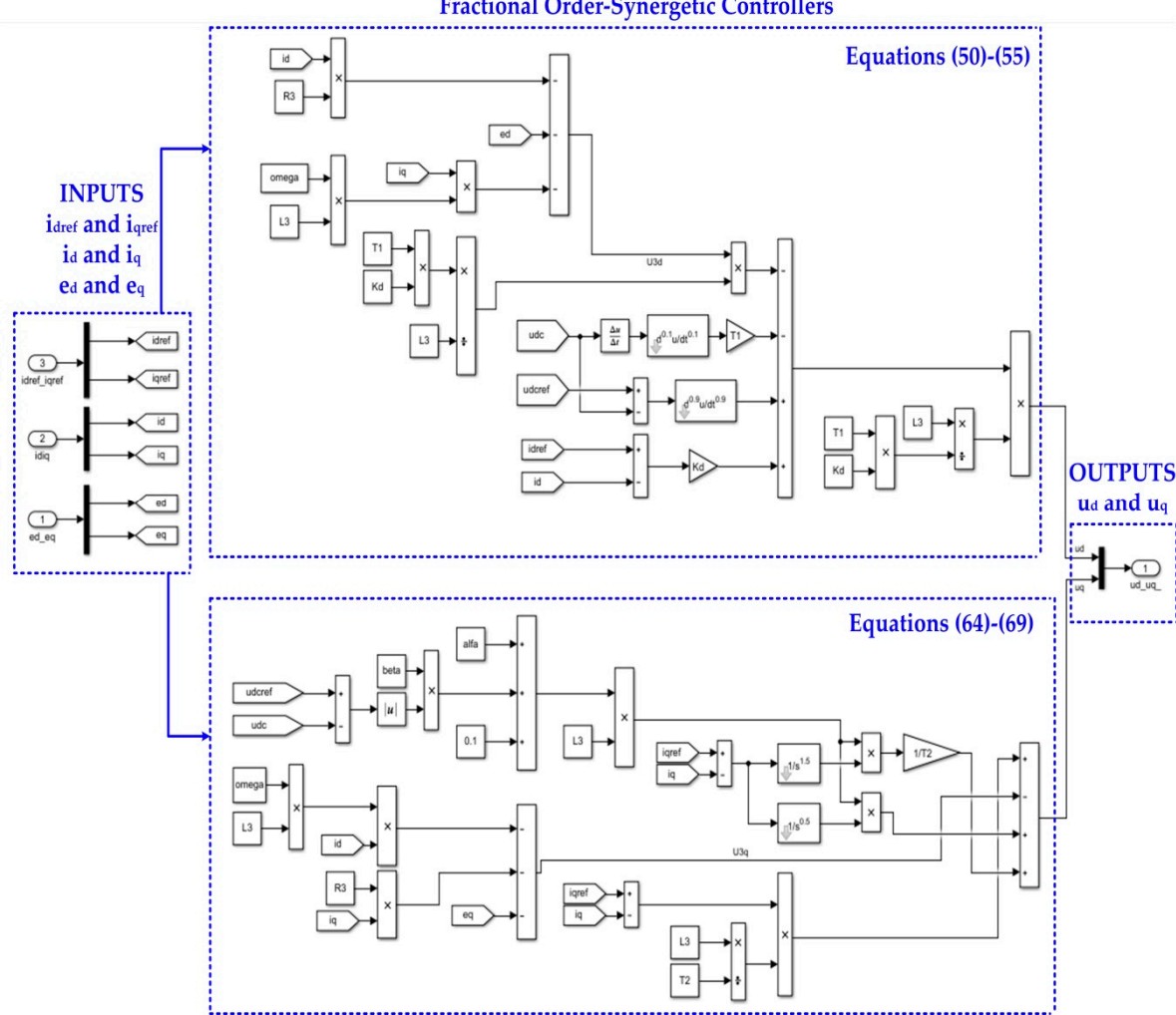

**Figure 6.** Matlab/Simulink implementation block diagram for FO-synergetic controllers.

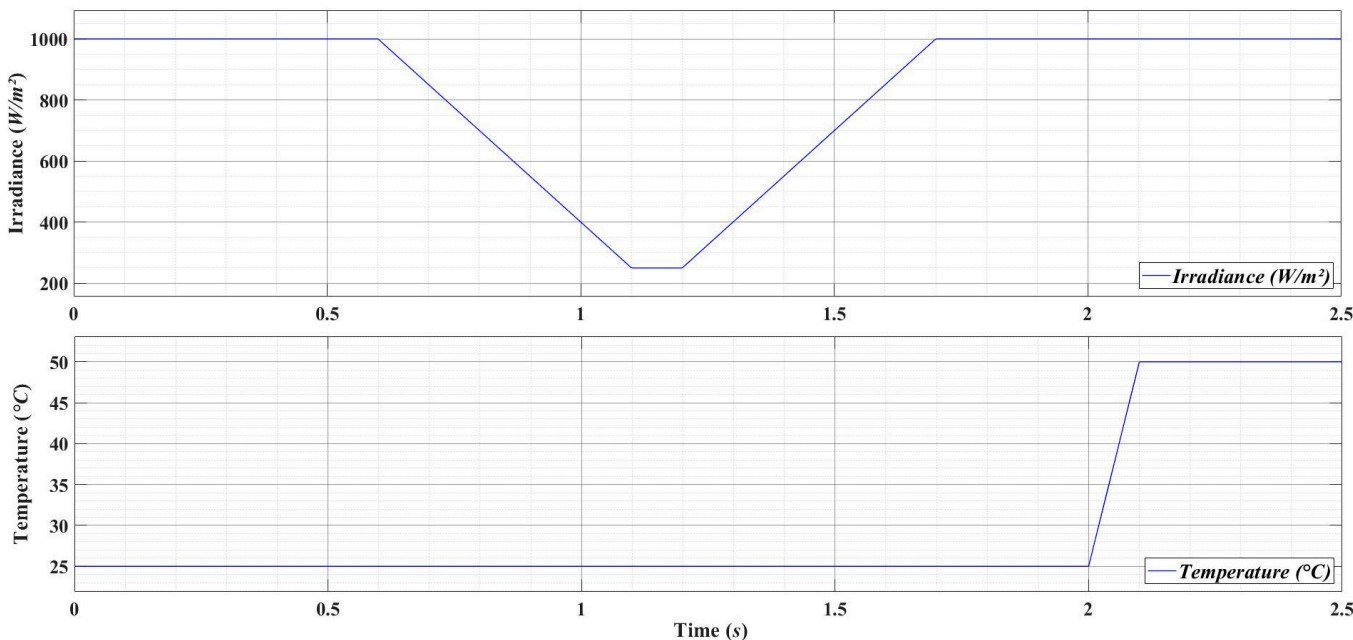

**Figure 7.** Time evolution of the irradiance and temperature signals type 1.

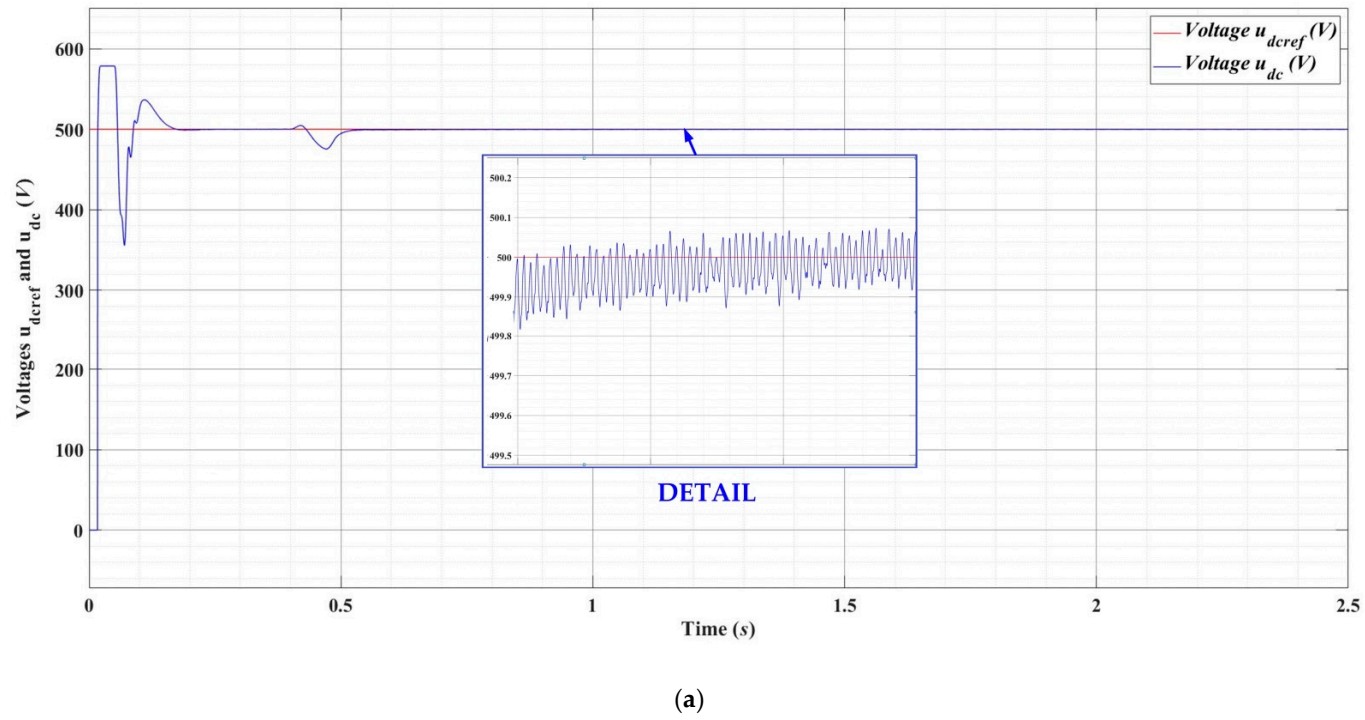

(**a**)

**Figure 8.** *Cont.*

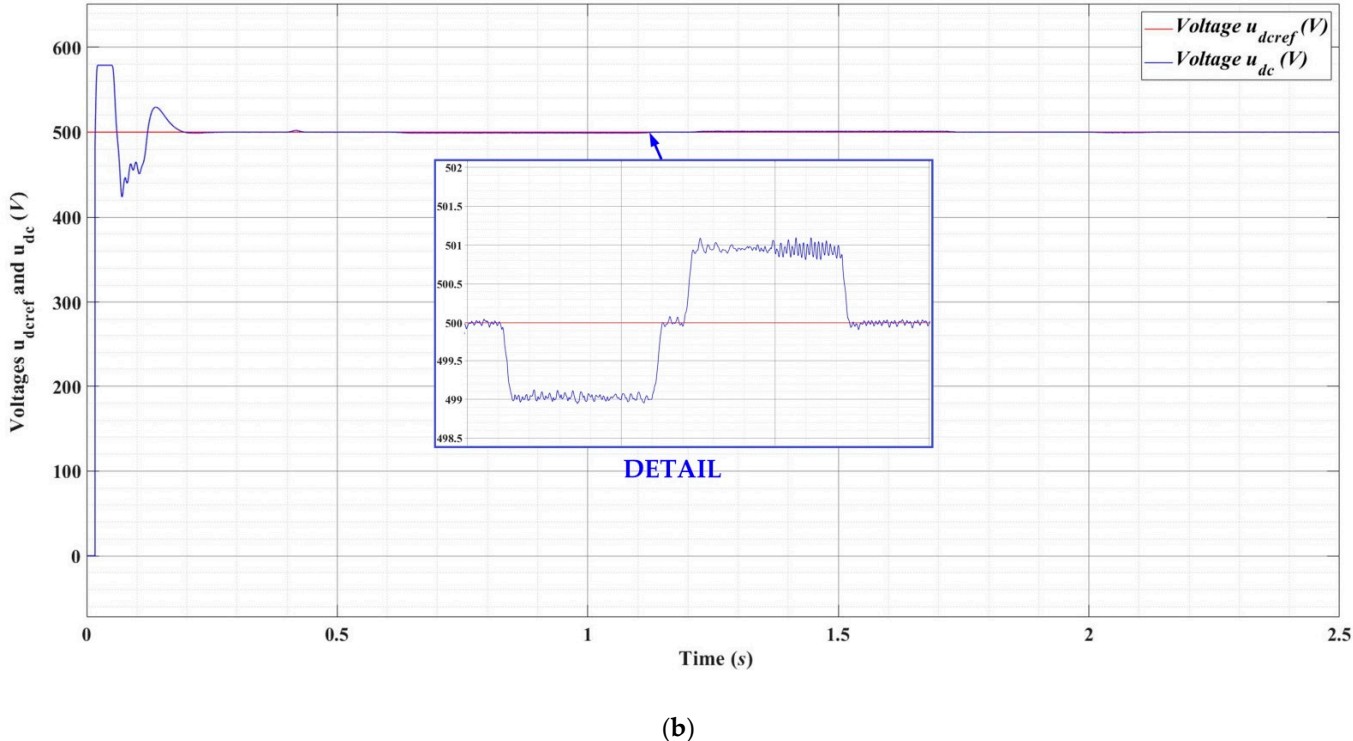

(**b**)

**Figure 8.** Time evolution of the $u_{dc}$ for the irradiance and temperature signals type 1, at 10 kvar load: (**a**) FO-SMC/FO-SYN controllers; (**b**) classical PI controllers.

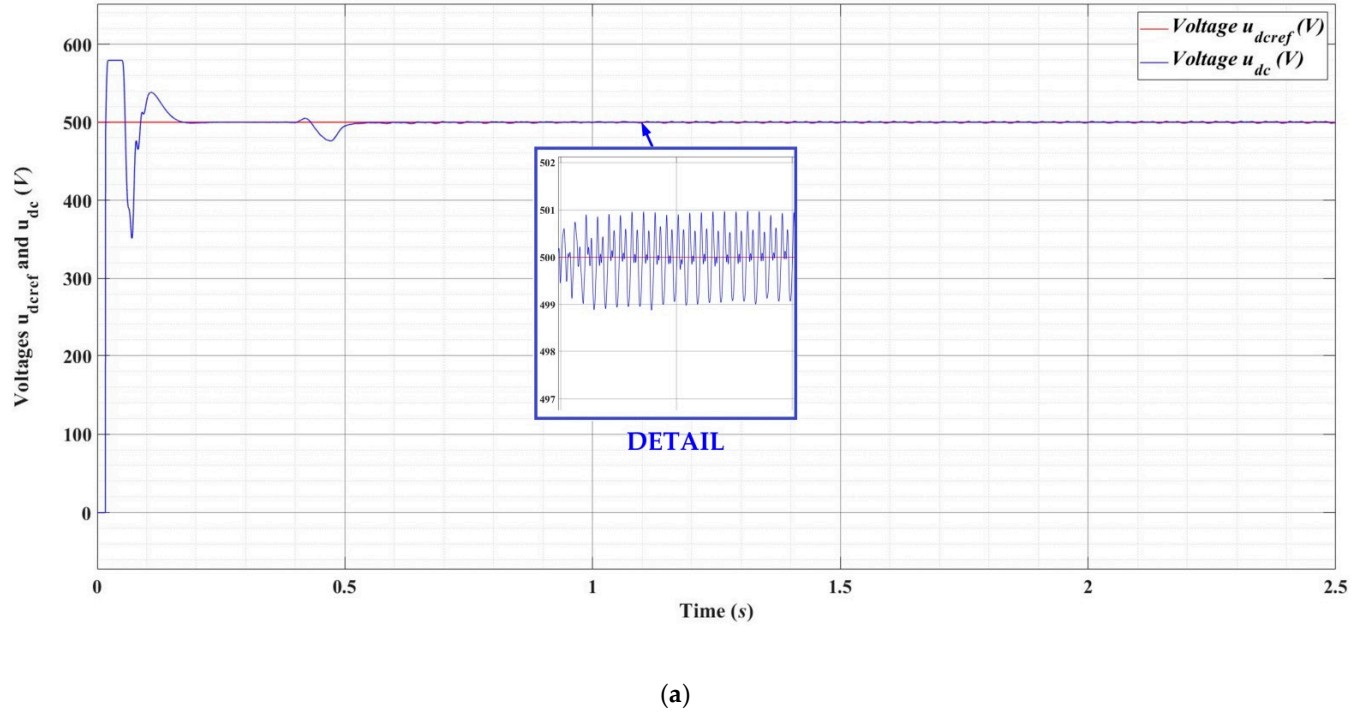

(**a**)

**Figure 9.** *Cont.*

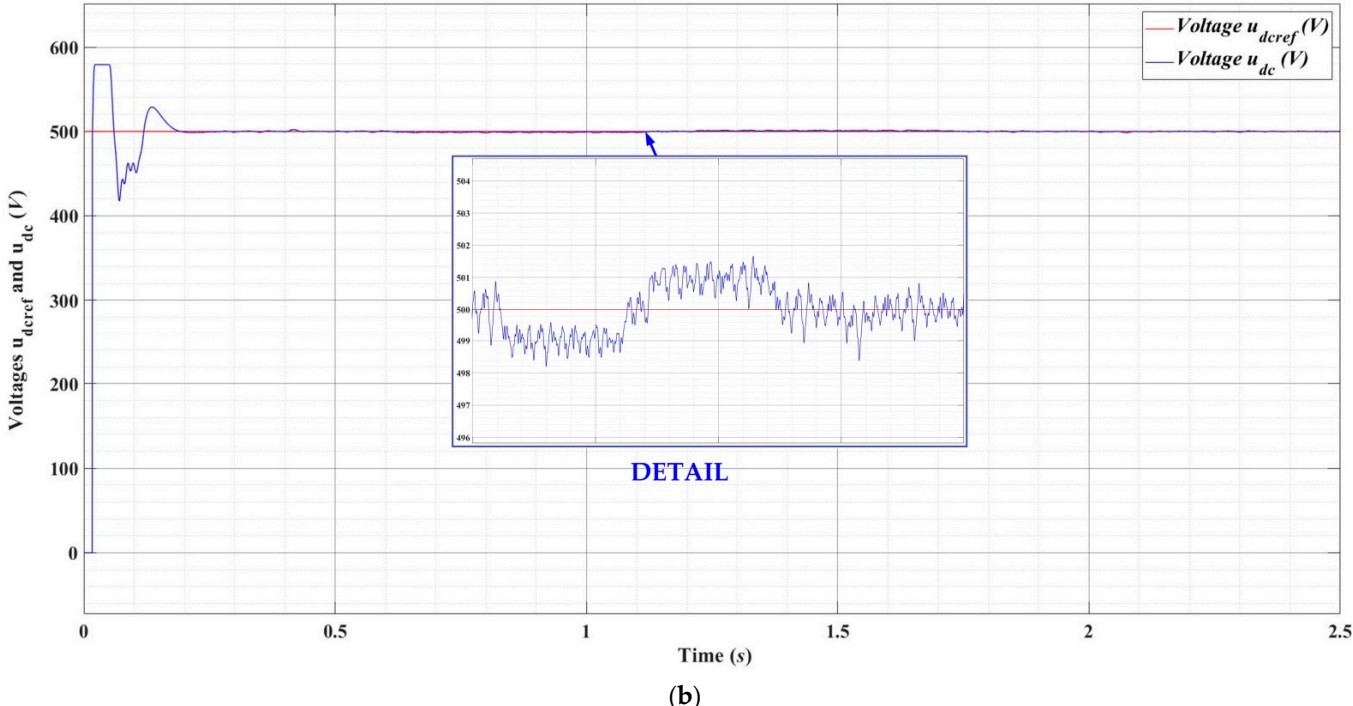

(**b**)

**Figure 9.** Time evolution of the $u_{dc}$ for the irradiance and temperature signals type 1, at 13 kvar load: (**a**) FO-SMC/FO-SYN controllers; (**b**) classical PI controllers.

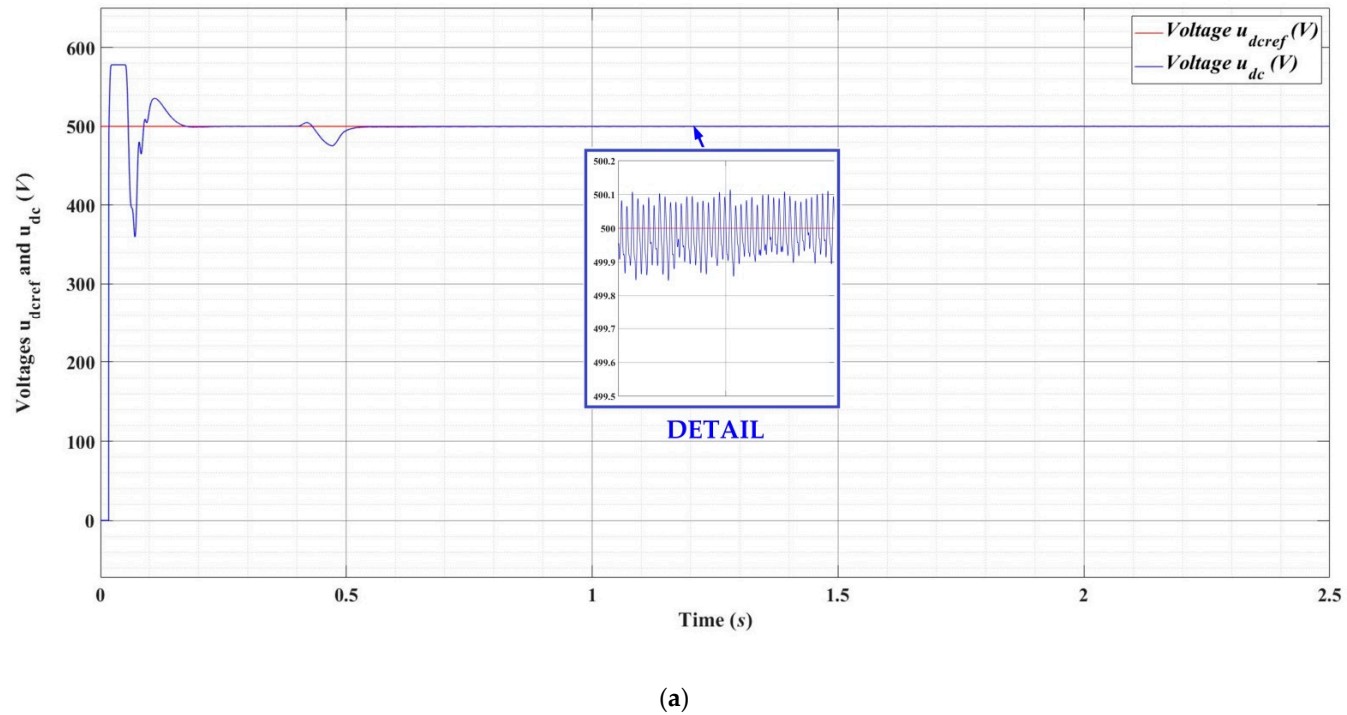

(**a**)

**Figure 10.** *Cont.*

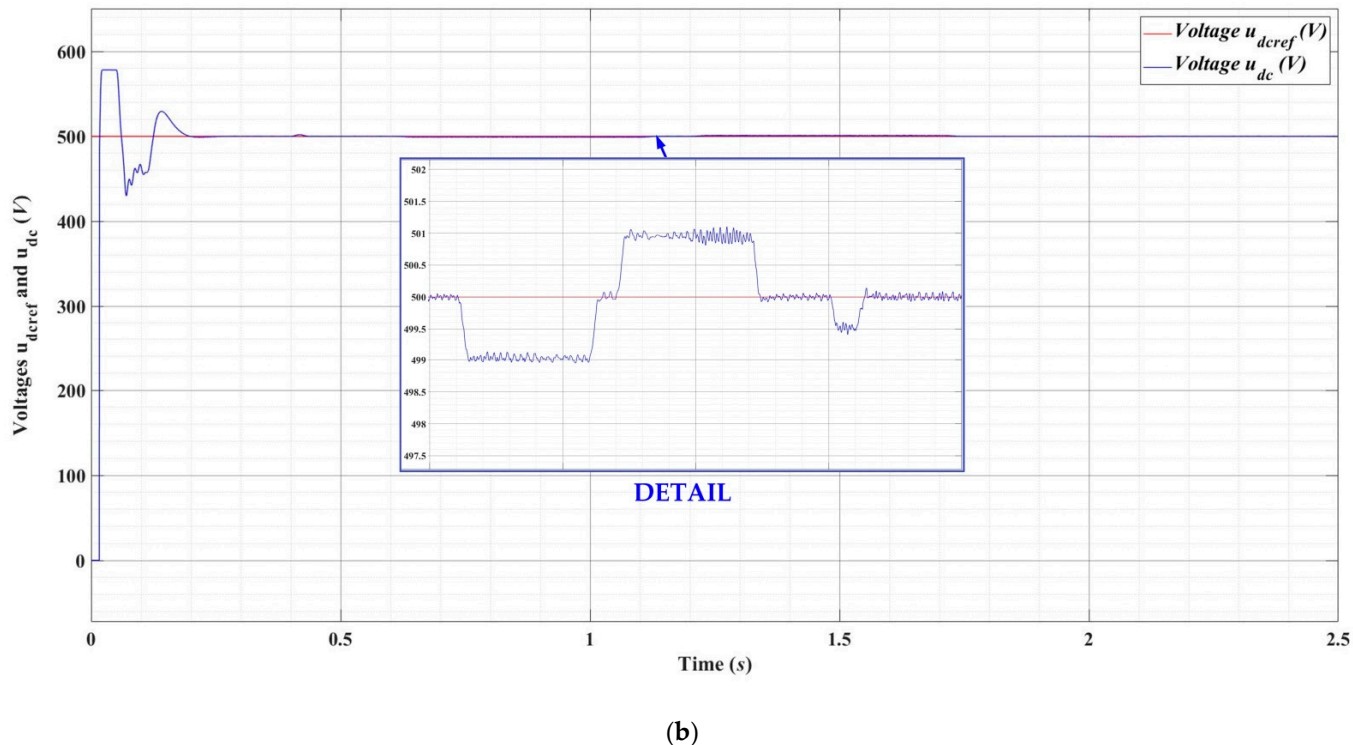

**(b)**

**Figure 10.** Time evolution of the $u_{dc}$ for the irradiance and temperature signals type 1, at 7 kvar load: (**a**) FO-SMC/FO-SYN controllers; (**b**) classical PI controllers.

Figure 8a shows the response of the FO-SMC type control system for the control of the DC voltage $u_{dc}$ combined with the FO-synergetic type control system for the control of currents $i_d$ and $i_q$ (FO-SMC/FO-SYN controllers), if the DC voltage reference $u_{dcref}$ = 500 V, and Figure 8b shows the response of the control system where the controllers used for both the control of the DC voltage $u_{dc}$ and for the control of currents $i_d$ and $i_q$ are of the PI type. After the validation of the control system start-up (after 50 ms), the MPPT algorithm start-up occurs (after 100 ms), and the end of the transitory regime (after 250 ms) is noted. In steady state, a steady-state error of 0.1 V, i.e., 0.02%, is noted for the FO-SMC/FO-SYN controllers, while the steady-state error for the PI controller is of 1 V, i.e., 0.2%. If the load is varied by a 30% increase or decrease, reaching 13 or 7 kvar, respectively, the superiority of the control of the grid-connected PV system based on FO-SMC/FO-SYN controllers is noted in Figures 9 and 10.

Regarding the dynamic regime, Figure 11 shows the response of the control of the grid-connected PV system based on FO-SMC/FO-SYN controllers as compared to PI controllers, where at time $t$ = 1 s, the reference signal $u_{dcref}$ undergoes a step variation at 550 V.

The parameters of the PI controller for $u_{dc}$ control are $K_p$ = 7 and $K_i$ = 800 and the parameters of the PI controller for $i_d$ and $i_q$ currents are $K_p$ = 0.3 and $K_i$ = 20 [15].

The parameters of the FO-SMC/FO-SYN controllers (presented in Section 3) are; $c_1$ = 100; $c_2$ = 1; $c_3$ = 1; $k$ = 180; $\varepsilon$ = 110; $C_2$ = 6000e-06; $T_1$ = 0.01; $T_2$ = 0.01; $K_p$ = 100; $K_d$ = 0.2; $L_3$ = 2.5000e-04; $R_3$ = 0.0019; $\omega$ = 2·π·60.

It is noted that the performances in the dynamic regime, as well as those in the stationary regime, are superior in the case of using FO-SMC/FO-SYN controllers, and in Figure 11, an override of 0.2% (1 V) and a response time of 20 ms are noted, compared to an override of 1% (5 V) and a response time of 50 ms in the case of PI controllers.

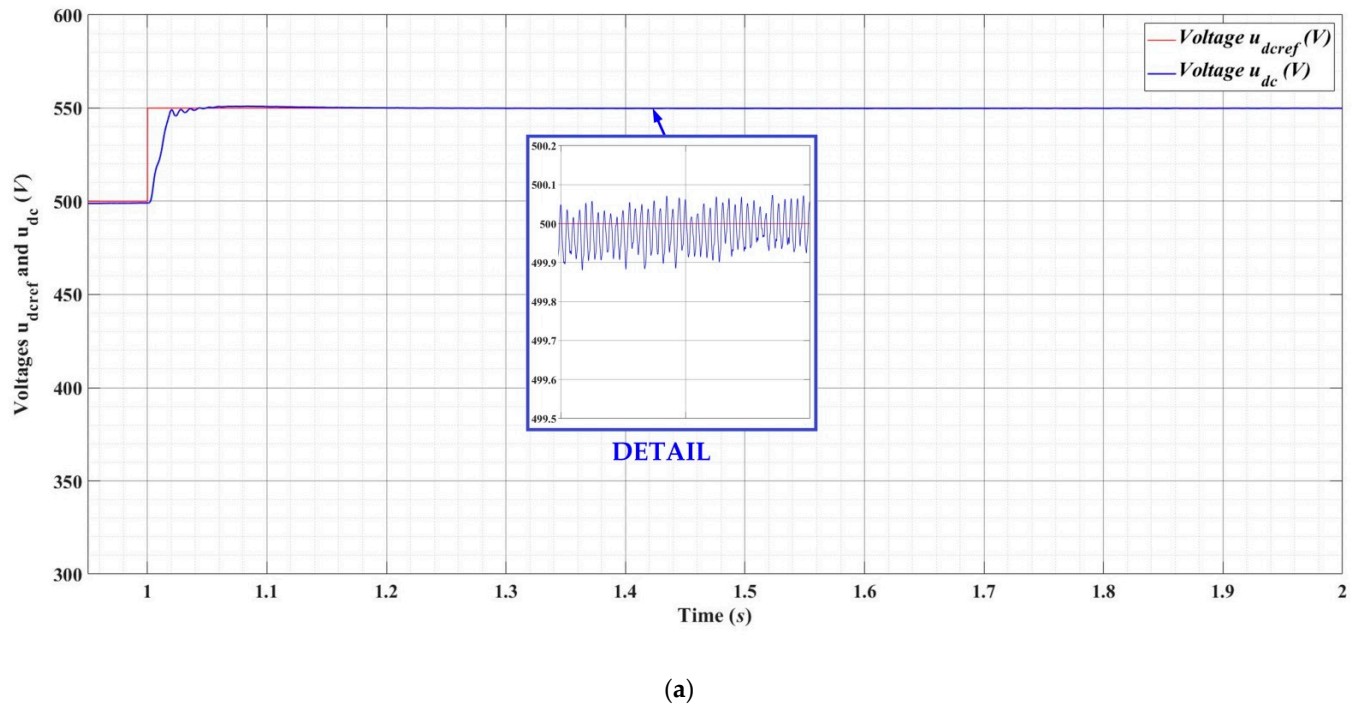

(**a**)

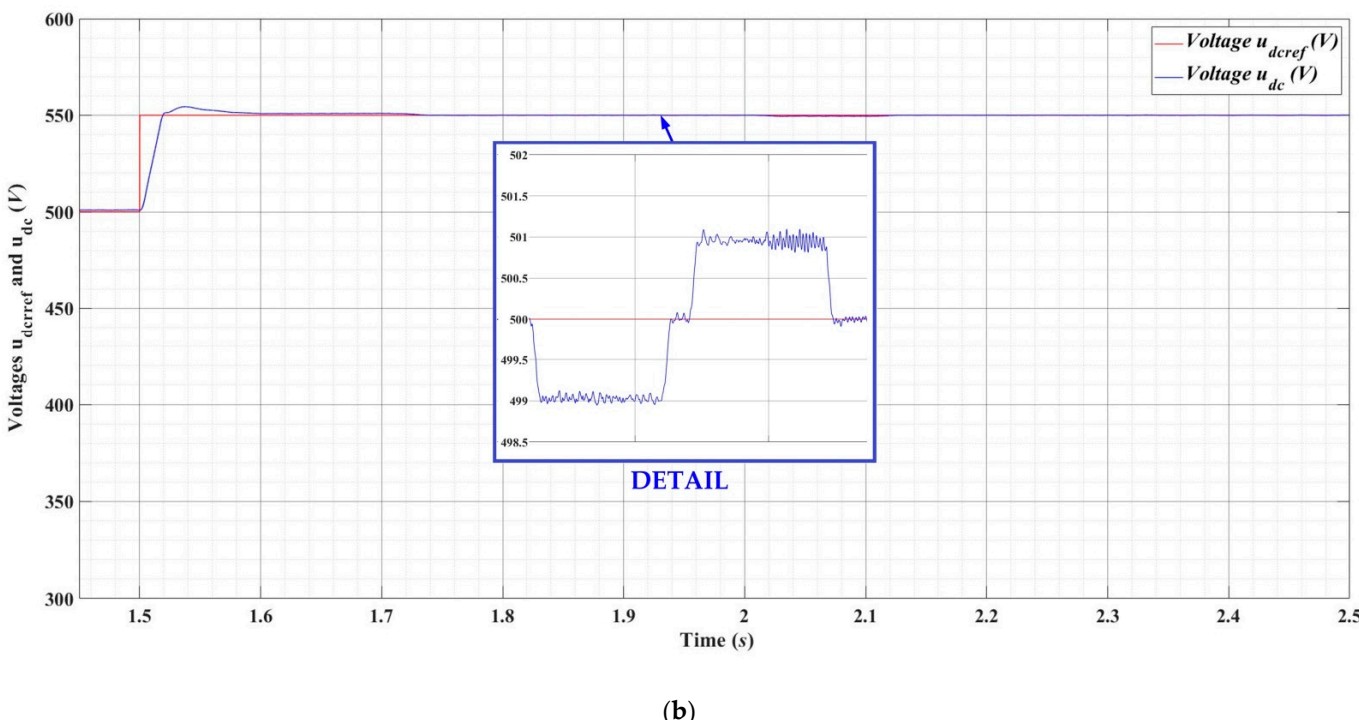

(**b**)

**Figure 11.** Time evolution of the $u_{dc}$ for the irradiance and temperature signals type 1, at 10 kvar load for a step variation of $u_{dcref}$ from 500 to 550 V: (**a**) FO-SMC/FO-SYN controllers; (**b**) classical PI controllers.

Figures 12–16 show a series of waveform graphs regarding the time evolution of the main inputs of interest in the control of the grid-connected PV system. Thus, Figure 12 shows the time evolution of $i_d$ and $i_q$ currents, where it is noted that $i_d$ follows the $i_{dref}$ reference provided by the FO-SMC controller, while $i_q$ follows the set reference $i_{qref} = 0$. Figure 13 shows the evolution of the average power and voltage of the PV array, $P_{mean}$ and $U_{mean}$. Further, with regard to the DC-DC converter, Figure 13 presents the time evolution of the duty cycle signal provided by the MPPT algorithm, and with regard to the DC-AC

converter, it presents the time evolution of the modulation index, a signal which is supplied by the control system of the VSC controller, which supplies control pulses. Figure 14 shows the evolution over time of the output voltage between two phases of the DC-AC converter. Figure 15 shows the time evolution of the voltage and current on a phase of the transformer for the connection to the main grid.

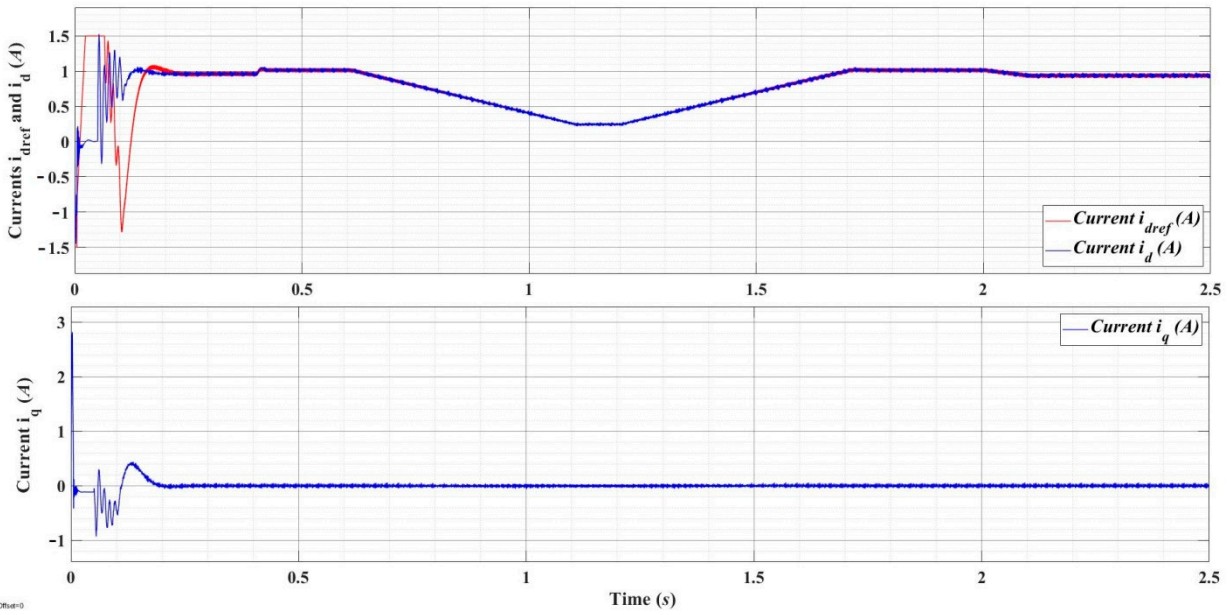

**Figure 12.** Time evolution of the $i_d$ and $i_q$ currents.

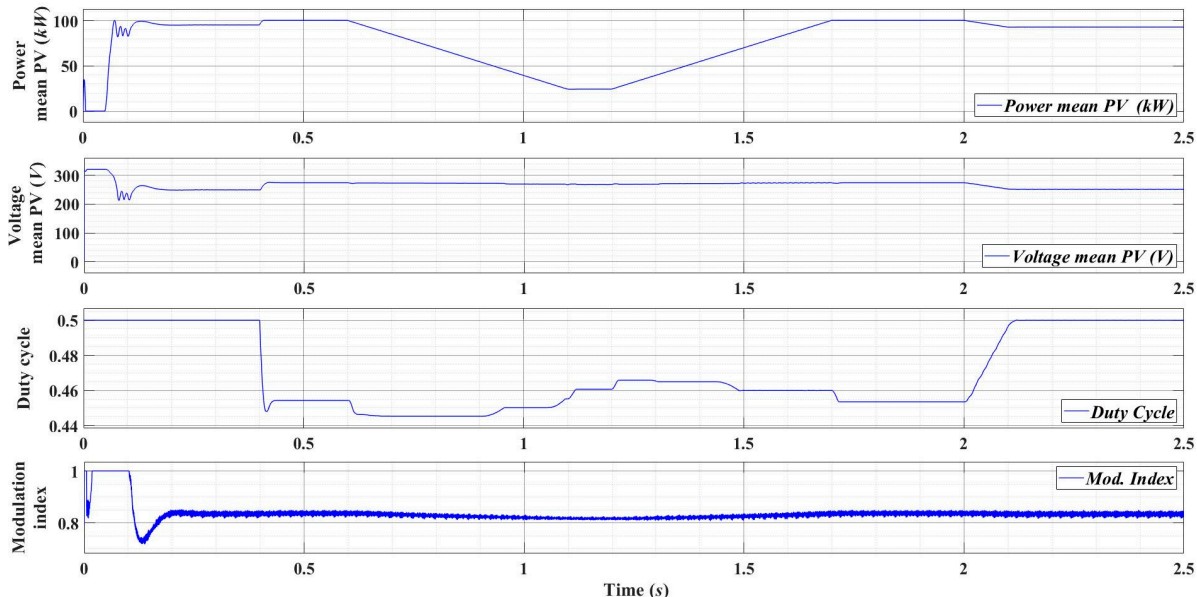

**Figure 13.** Time evolutions of the power $P_{mean}$ and voltage $U_{mean}$ of the PV, duty cycle of the DC-DC converter, and modulation index of the DC-AC converter.

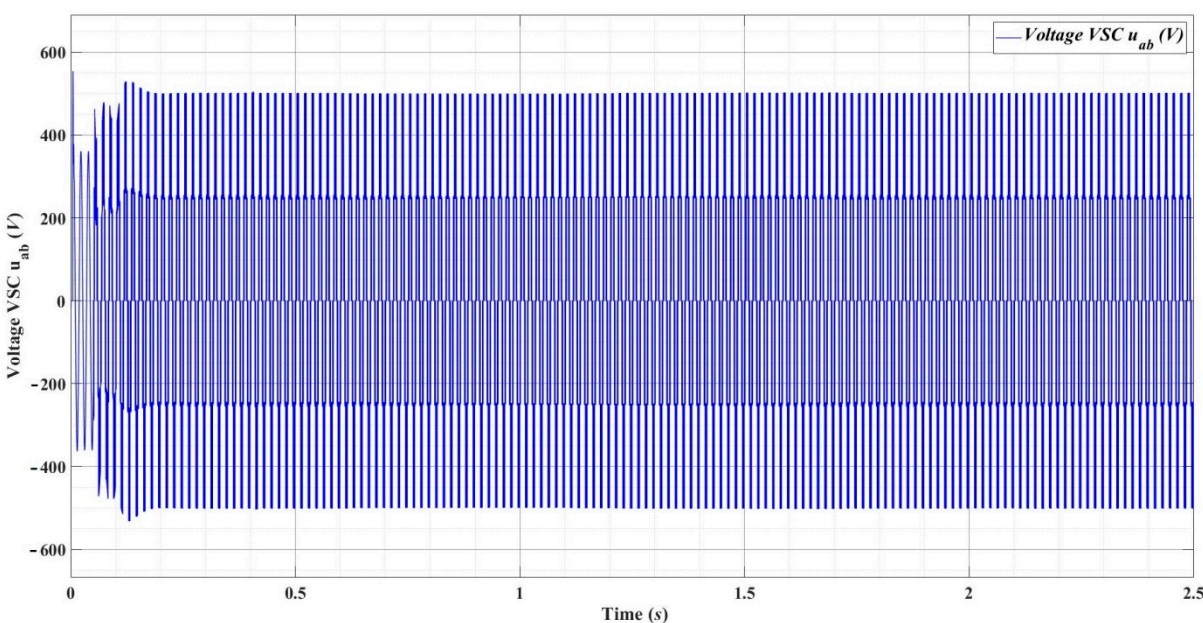

**Figure 14.** Time evolution of the voltage $u_{ab}$ of the voltage source converter (VSC) controller.

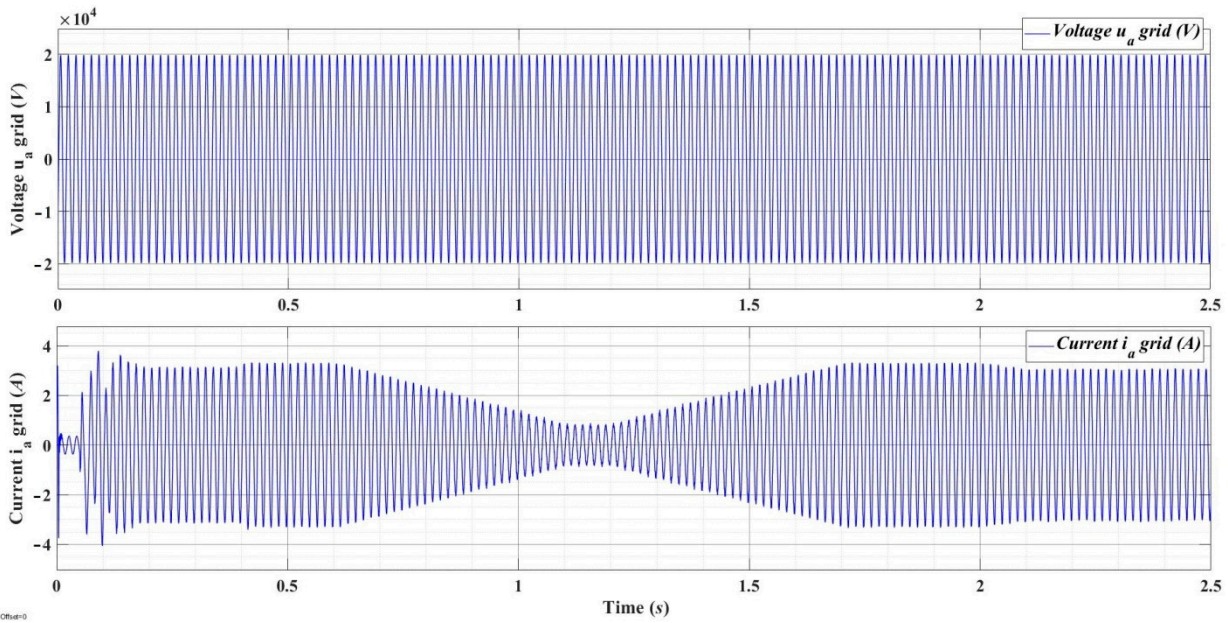

**Figure 15.** Time evolution of the voltage $u_a$ and current $i_a$ of the main grid.

Moreover, Figure 16 shows the time evolution of the active power which flows between the analyzed system and the main grid.

The control of the grid-connected PV system implemented in [15] is also discussed in [26], in which the control system of $u_{dc}$ voltage is of the SMC type, the control systems of currents $i_d$ and $i_q$ are of the classic PI type, and the time evolution of the irradiance and temperature signals is presented in Figure 17. For the FO-SMC/FO-SYN controllers proposed in this article, Figures 18–20 show the time evolution of the voltage $u_{dc}$ in the DC intermediate circuit, if the reference voltage $u_{dcref}$ is of 500 V. Superior performances are also noted in this case, both for the nominal load of 10 kvar and for its variations to 13 and 7 kvar. It is noted that the steady-state error is maintained at 0.1 V (0.02%).

Due to the fact that the basic model in Matlab/Simulink is complex and has all the aspects regarding the transformation chain from the PV array to the main grid connection

and considering that it is used for comparison in other papers [15,26,27,31], this model can be considered as a benchmark for the control of the grid-connected PV system. Thus, the superior performance obtained by using the FO-SMC/FO-SYN controllers proposed in this article can be considered as a validation of the proposed control system.

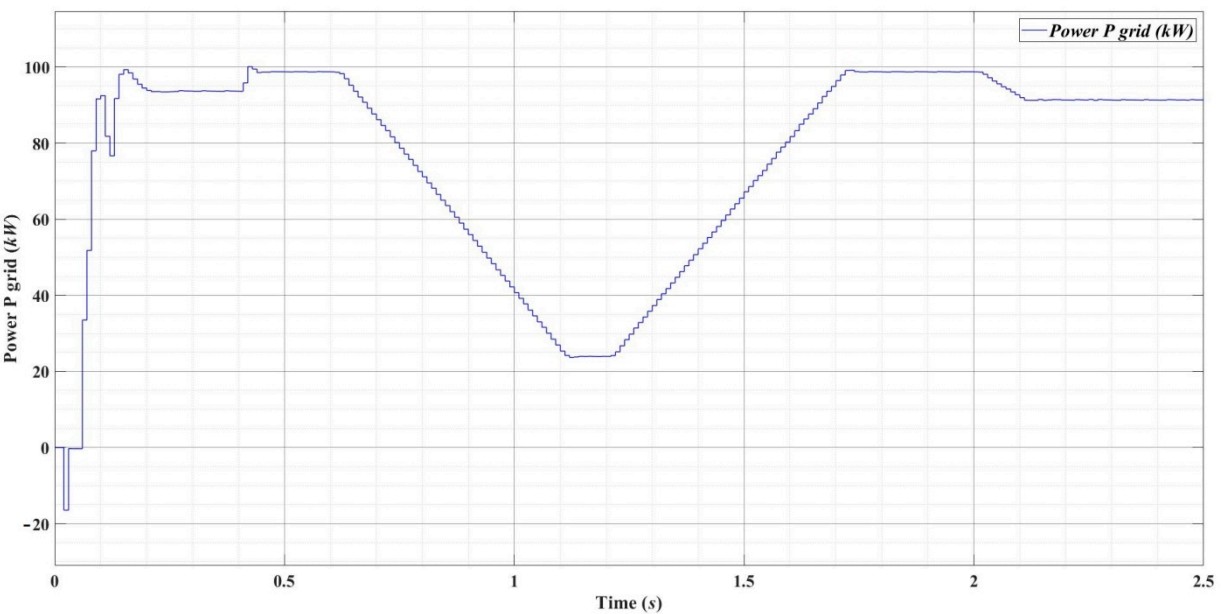

**Figure 16.** Time evolution of the power flow P between PV system and main grid.

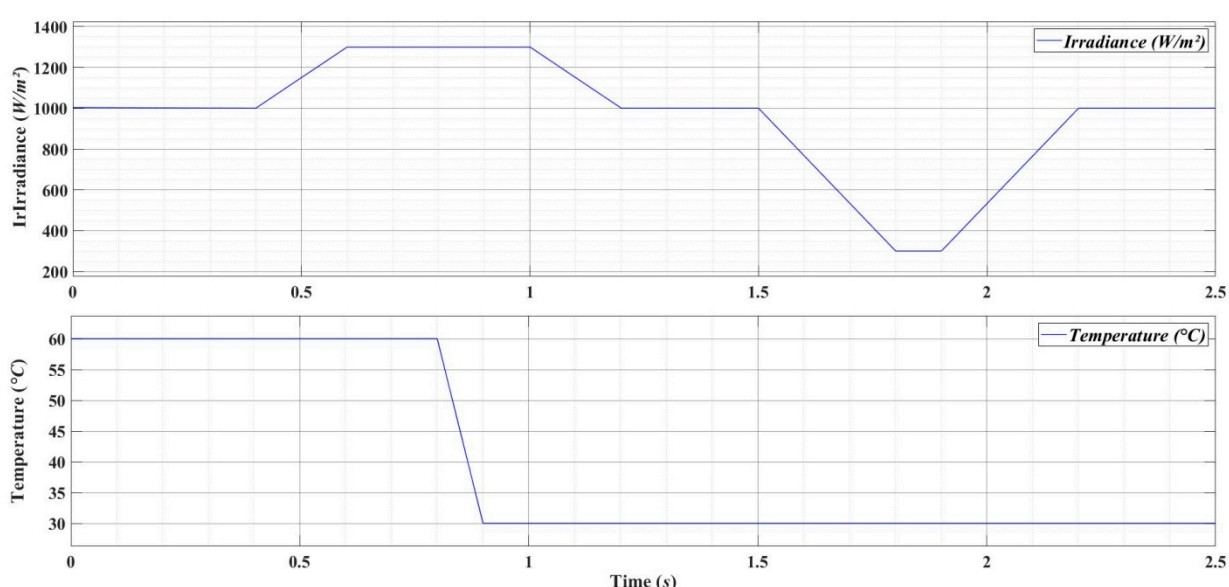

**Figure 17.** Time evolution of the irradiance and temperature signals type 2.

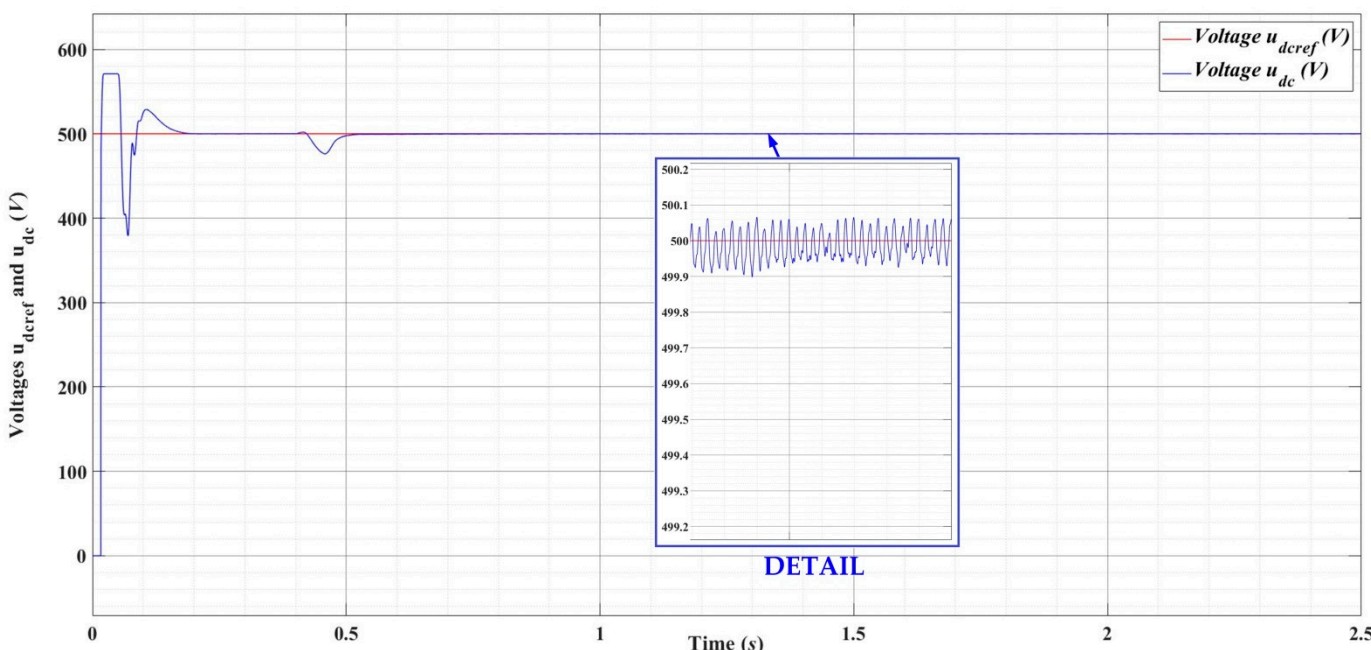

**Figure 18.** Time evolution of the $u_{dc}$ for the irradiance and temperature signals type 2, at 10 kvar load with FO-SMC/FO-SYN controllers.

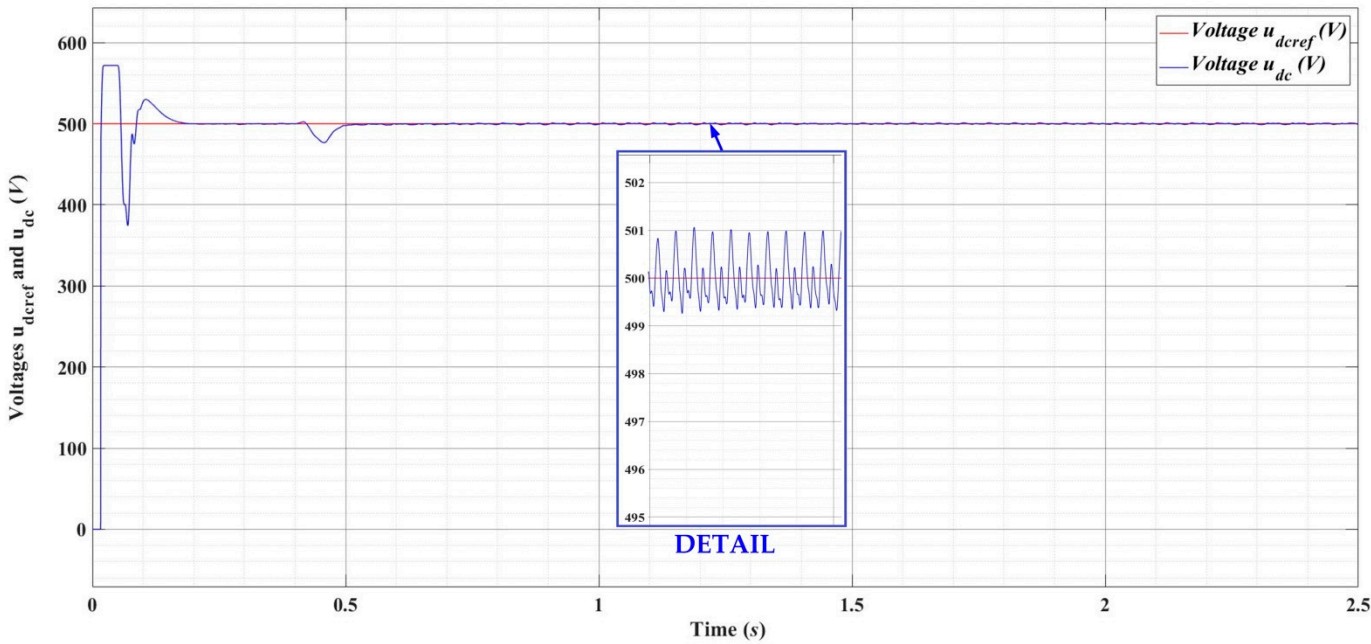

**Figure 19.** Time evolution of the $u_{dc}$ for the irradiance and temperature signals type 2, at 13 kvar load with FO-SMC/FO-SYN controllers.

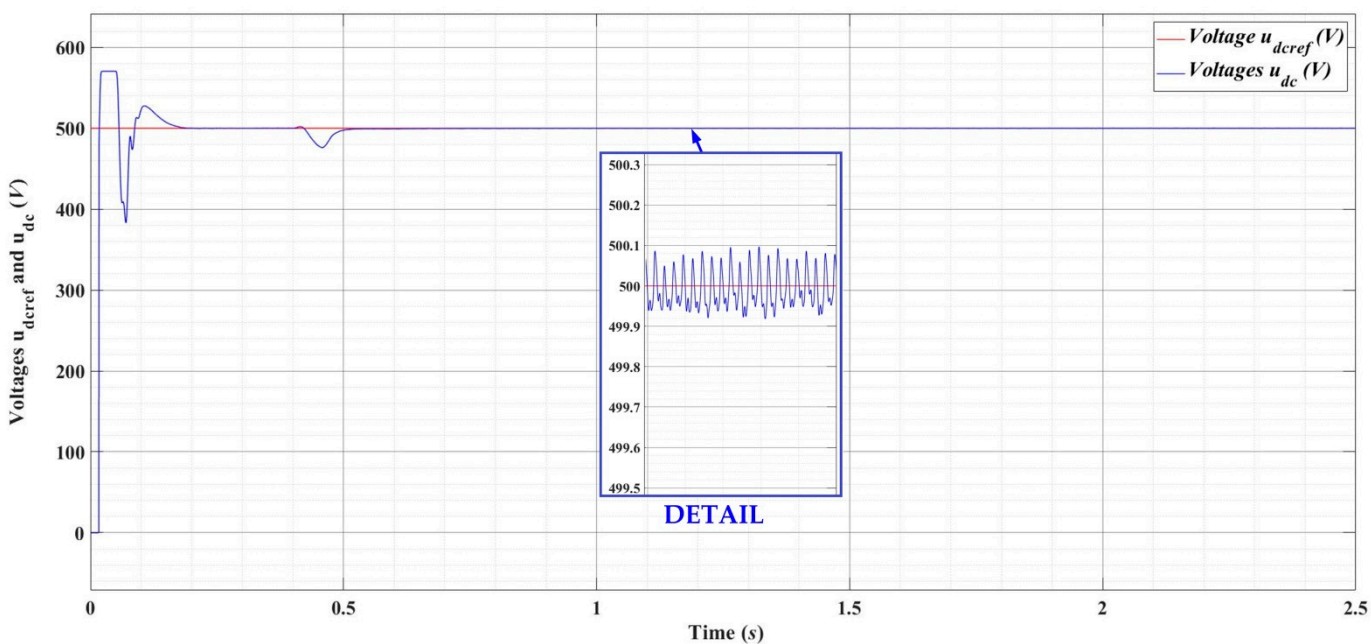

**Figure 20.** Time evolution of the $u_{dc}$ for the irradiance and temperature signals type 2, at 7 kvar load with FO-SMC/FO-SYN controllers.

## 5. Conclusions

This article presents the control of a grid-connected PV system using FO-SMC and FO-synergetic controllers. The mathematical model of a PV system connected to the main grid is presented together with the chain of intermediate elements: the DC-DC boost converter, the DC intermediate circuit, the DC-AC converter, the filtering block, and the transformer for the connection to the main grid, together with their control systems. The robustness and superior performance of an SMC-type controller for the control of $u_{dc}$ voltage in the DC intermediate circuit are combined with the advantages provided by the flexibility of using synergetic control for the control of currents $i_d$ and $i_q$. In addition, these control techniques are suitable for the control of nonlinear systems, and it is not necessary to linearize the controlled system around a static operating point; thus, the control system achieved is robust to parametric variations and provides the required static and dynamic performance. Further, by approaching the synthesis of these controllers using the fractional calculus for integration and differentiation operators, this article proposes a control system based on FO-SMC/FO-SYN controllers. The validation of the synthesis of the proposed control system is achieved by comparing it with a benchmark for the control of a grid-connected PV system implemented in Matlab/Simulink.

**Author Contributions:** Conceptualization, M.N.; data curation, M.N. and C.-I.N.; formal analysis, M.N. and C.-I.N.; funding acquisition, M.N.; investigation, M.N. and C.-I.N.; methodology, M.N. and C.-I.N.; project administration, M.N.; resources, M.N. and C.-I.N.; software, M.N. and C.-I.N.; supervision, M.N.; validation, M.N. and C.-I.N.; visualization, M.N. and C.-I.N.; writing—original draft, M.N. and C.-I.N.; writing—review and editing, M.N. and C.-I.N. All authors have read and agreed to the published version of the manuscript.

**Funding:** The paper was developed with funds from the Ministry of Education and Scientific Research—Romania as part of the NUCLEU Program: PN 19 38 01 03.

**Institutional Review Board Statement:** Not applicable.

**Informed Consent Statement:** Not applicable.

**Data Availability Statement:** Data sharing not applicable.

**Conflicts of Interest:** The authors declare no conflict of interest.

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
