# Peer review of "Fractional-Order Control of Grid-Connected Photovoltaic System Based on Synergetic and Sliding Mode Controllers"

_energies, doi:10.3390/en14020510_

Round 1

Reviewer 1 Report

The robustness of the DC/AC converters is important for its functioning. Grid connected photovoltaic systems are important in the mix of energy sources. I think the article is suitable for Electronics journal. The paper is well written as well, however some concern are existed for this reviewer.

The Abstract in its current from is an alternative Introduction, it should clearly describe the scope with more focusing on the proposed approach and results of the study. Thus, the structure of the abstract need to be changed covering, overview, problem, methods, results, conclusion.

Several controllers are presented in literatures based on PI and Passivity theory. What is the main contribution of your proposed method over the other existing techniques?

Introduction section should provide a critical analysis of the available and appropriate literature to identify an objective whose accomplishment will provide a significant contribution to the field. The research gap in the literature should be clearly exposed.

Literature review on control of DC/AC converter for grid connected photovoltaic systems should be improved. Thus, the authors are invited to update the introduction and refer the following reference in the literature review:  (i) Nonlinear Voltage Control for Three-Phase DC-AC Converters in Hybrid Systems: An Application of the PI-PBC Method, Electronics 9 (5), 847.

Why do you choose an LC-filter for this application instead of a simple L-filter? Explain it in more details. How were the L and R parameters of this filter chosen?

Please clarify what switching frequency has been selected and its impact.

Author Response

Dear reviewer, thanks for your recommendations.

1) We have added to the Introduction as per to your instructions.

2) A type of special controllers used for the control of linear and non-linear systems is based on PI and Passivity theory [25]. This type of control, known in the beginning as the hyperstability control theory is based on the appropriate description of the system in the closed loop in the form of the Lagrangian or the port-controlled Hamiltonian, which defines the behavior of the system through energy functions. This description is generally expressed in the form of solutions to partial differential equations, which require an increased degree of difficulty in the implementation of these types of controllers.  

            We write the system in the form of the model Hamiltonian port:

xdot=[J-R][dH(x)/dx]+gu+ζ

where the state vector is x, J, and R are the interconnection and damping matrices, respectively, H(x) represents the total energy stored in the system, g is the input matrix, u is the control input vector, and z represents the external input.

As a parallel to the stability of non-linear systems based on Lyapunov stability, we select the Lyapunov function of the type V(x,z)=H(x)+(1/2)+zTKz, where H(x) is the Hamiltonian function, and the second term is a quadratic expression as a function of the integral variables that deal with these new stable variables introduced by the PI-PBC method.

            Chronologically speaking, the passivity theory was introduced in engineering field in 1965 by Vasile Mihai POPOV, the greatest Romanian scientist in automation, and it was known as the hyperstability theory (POPOV hyperstability theory, 1965).

Such an implementation is presented in the paper “Serra, F.M.; Fernández, L.M.; Montoya, O.D.; Gil-González, W.; Hernández, J.C. Nonlinear Voltage Control for Three-Phase DC-AC Converters in Hybrid Systems: An Application of the PI-PBC Method. Electronics 20209, 847”. While not being a direct comparison between the results obtained using the algorithm proposed in our article and the above article, the value of the steady-state error is of 0.1V for our paper, compared to about 1V in the mentioned article.

3) In this article, a control system is proposed, which proves to be superior to other control systems proposed in other articles in which the implementation scheme is the same as that in Figure 4, and presented in the form of a benchmark in Matlab/Simulink [15]. The components which ensure the connection of the PV system to the main grid are: the DC-DC Boost converter, the DC intermediate circuit, the DC-AC converter, the filtering block, and the transformer for the connection to the main grid, together with their control systems. The logic chain of the components in Figure 4 is:

SIMULATED INPUTS FOR PV --> PV ARRAY --> MPPT ALGHORITM --> DC-DC BOOST CONVERTER --> VOLTAGE AND CURRENT MEASUREMENT FOR DC INTERMEDIATE CIRCUIT --> CONTROL ALGORITHM --> DC-AC --> LOAD AND CONNECTION TO POWER GRID (including LC Filter) --> POWER GRID.

In order to maintain the same comparison conditions when comparing the control algorithms, all other elements were maintained, including the LC Filter.

4) In the Matlab/Simulink implementation, the sample time used is of one microsecond for the PWM generator command signals for the DC-DC and DC-AC converters. For the control system of the voltage and currents, but also for the PLL type synchronization loop, the sampling period is of 0.1ms. In summary, the running frequency of the control algorithm is of 10kHZ, and for the PWM-type actuators is of 1MHz [15]. These frequencies fall within the usual frequencies for Matlab/Simulink implementation, but also the real-time running frequencies [32].

Reviewer 2 Report

1) References linked with Fractional Order Control must be included. For example the authors can check the theory presented in [A1].

[A1].- https://doi.org/10.1109/ICMSAO.2019.8880408

2) Justification of eq. (29) must be clarified. How is possible to replace he sgn() function by eq. (29) ?.

3) MPPT presented in [15] must be clarified. For example, in steady-state common MPPT based on Perturb and Observe MPPT has a three level fashion. How is ensured the MPPT on your system?

4) Fig. 5 and Fig. 6 must be summarized or eliminated. The information provided by these figures is minor.

5) Experimental results are requested to validate the proposed control technique.

Author Response

Dear reviewer, thanks for your recommendations.

1) We have added the article recommended by you to the list of references: “Mehiri, A.; Bettayeb, M.; Hamid, A. Fractional Nonlinear Synergetic Control for Three Phase Inverter Tied to PV System. In Proceedings of the 8th International Conference on Modeling Simulation and Applied Optimization (ICMSAO), Manama, Bahrain, 15–17 April 2019, pp. 1–5.”. This article presents such an implementation in which the classic PI-type controllers are replaced with FO-Synergetic controllers.

Superior results are obtained by using the control structure proposed in this article, as well as the proposed macro-variables for the synthesis of control laws.

In the article previously mentioned, a single surface/macro-variable is used:

? = ???+ ??Ι??.

The following macrovariables are used in our paper:

The surface S: (see the attached)  for FO-SMC control law

The macrovariables: (see the attached) and (see the attached) for FO-Synergetic control laws.

While not being a direct comparison between the results obtained using the algorithm proposed in our article and the above article, the value of the steady-state error is of 0.1V for our paper, compared to about 1V in the mentioned article.     

2) Among the disadvantages of the SMC type control we mention the occurrence of the chattering phenomenon, which represents the occurrence of oscillations in the control input due to the process of synthesis of the controller. For this purpose, to reduce oscillations, transition functions smoother than the sgn function are used (between conventional thresholds +1 and -1) followed by a corresponding filtration.

Following [32], to improve convergence and reduce high frequency oscillations, the sgn function (blue line in the figure represented below) is replaced with the function below:

 (red line in the figure represented below)  

For a=4 and b=0, , and a smoothed transition is achieved between -1 and 1 (see the attached).

4) Figure 5 and 6 present the Matlab/Simulink implementation block diagram for the FO-SMC Controller and the FO-Synergetic Controller. In our opinion, the figures should be kept because they summarize the manner of implementation of the proposed control algorithm and simplify the reader’s attempt to implement this algorithm himself, considering that it summarizes precisely the equations and parameter values required for the implementation in Matlab/Simulink.

3) and 5) Switching duty cycle is optimized by a MPPT controller that uses the ‘Incremental Conductance + Integral Regulator’ technique. This MPPT system automatically varies the duty cycle in order to generate the required voltage to extract the maximum power [15].

In this article, a control system is proposed, which proves to be superior to other control systems proposed in other articles, in which the implementation scheme is the same as that in Figure 4, and presented in the form of a benchmark in Matlab/Simulink [15]. The components which ensure the connection of the PV system to the main grid are: the DC-DC Boost converter, the DC intermediate circuit, the DC-AC converter, the filtering block, the transformer for the connection to the main grid, together with their control systems. The logic chain of the components in Figure 4 is:

SIMULATED INPUTS FOR PV --> PV ARRAY --> MPPT ALGHORITM --> DC-DC BOOST CONVERTER --> VOLTAGE AND CURRENT MEASUREMENT FOR DC INTERMEDIATE CIRCUIT --> CONTROL ALGORITHM --> DC-AC --> LOAD AND CONNECTION TO POWER GRID --> POWER GRID.

In order to maintain the same comparison conditions when comparing the control algorithms, all other elements were maintained, including the MPPT Alghoritm. The comparisons were carried out just through numerical simulations in the Matlab/Simulink environment (both in our article and in the other articles).

In [32], the paper by the same authors of this paper, for the control of a PMSM using the same proposed control structure based on FO-SMC and FO-Synergetic controllers, superior results are obtained both by numerical simulations and by implementation in practice (in comparison with other articles – same condition for comparison and same criterion for comparison). A quote from [32] shows the superiority of the proposed control algorithm: “The PMSM used in these numerical simulations is implemented in the Power Systems/Simscape Electrical toolbox from Simulink, and in many scientific papers it is used as a benchmark, and, to our best knowledge, the settling time of 0.92 ms obtained when using the FO-SMC speed controller and the FO-synergetic current controller is the best settling time obtained for a usual range of the speed reference and load torque, and for any other proposed controllers.”

Round 2

Reviewer 1 Report

I consider that my questions were addressed appropriately. The paper can be published as it is.